# Mitochondrial prohibitin complex regulates fungal virulence via ATG24-assisted mitophagy

Yaqin Yan[1,2], Jintian Tang[3], Qinfeng Yuan[1], Caiyun Liu[1], Xiaolin Chen[1], Hao Liu[1], Junbin Huang[1], Chonglai Bao[2], Tom Hsiang [4] & Lu Zheng [1✉]

Prohibitins are highly conserved eukaryotic proteins in mitochondria that function in various cellular processes. The roles of prohibitins in fungal virulence and their regulatory mechanisms are still unknown. Here, we identified the prohibitins ChPhb1 and ChPhb2 in a plant pathogenic fungus *Colletotrichum higginsianum* and investigated their roles in the virulence of this anthracnose fungus attacking crucifers. We demonstrate that ChPhb1 and ChPhb2 are required for the proper functioning of mitochondria, mitophagy and virulence. ChPhb1 and ChPhb2 interact with the autophagy-related protein ChATG24 in mitochondria, and ChATG24 shares similar functions with these proteins in mitophagy and virulence, suggesting that ChATG24 is involved in prohibitin-dependent mitophagy. ChPhb1 and ChPhb2 modulate the translocation of ChATG24 into mitochondria during mitophagy. The role of ChATG24 in mitophagy is further confirmed to be conserved in plant pathogenic fungi. Our study presents that prohibitins regulate fungal virulence by mediating ATG24-assisted mitophagy.

[1] State Key Laboratory of Agricultural Microbiology/Hubei Key Laboratory of Plant Pathology, Huazhong Agricultural University, Wuhan 430070, China. [2] Institute of Vegetables Research, Zhejiang Academy of Agricultural Sciences, Hangzhou 310021, China. [3] Zhejiang Provincial Key Laboratory of Biometrology and Inspection & Quarantine, College of Life Sciences, China Jiliang University, Hangzhou 310018, China. [4] School of Environmental Sciences, University of Guelph, Guelph, ON N1G 2W1, Canada. ✉email: luzheng@mail.hzau.edu.cn

**M**itochondria are semi-autonomous double membrane-bound organelles that generate most of the adenosine triphosphate (ATP) required for diverse cellular

**Fig. 1 Domain prediction and phylogenetic analysis of prohibitin proteins in *C. higginsianum*. a** Visualization of protein domain structures of ChPhb1 and ChPhb2 annotated by DOG (Domain Graph, version 1.0) software. **b** Phylogenetic analysis of amino acid sequences of ChPhb1 and ChPhb2 with homologs from other fungal species. All the sequences were downloaded from NCBI database and their accession numbers are shown following the gene names. The numbers at branch nodes are bootstrap percentages out of 2000 replicates and the neighbor-joining method was used in phylogenetic tree construction. **c** Expression profiles of *ChPhb1* and *ChPhb2* relative to actin in conidia (Co) and hyphae (Hy) from PDB cultures and at different stages of infection on *Arabidopsis* (24, 48 and 72 hpi), as revealed by qRT-PCR. Error bars represent ± SD.

functions and are crucial to various physiological processes, including lipid metabolism, redox signaling, calcium and iron homeostasis, and programmed cell death[1,2]. Prohibitin proteins, which belong to the SPFH (stomatin/prohibitin/flotillin/HflKC) family, are ubiquitously expressed and highly conserved in eukaryotes[3,4]. The prohibitin complex contains two subunits, Phb1 and Phb2, which physically associate with each other to form a high-molecular-weight complex in the inner mitochondrial membrane[5,6].

The prohibitin complex plays a variety of roles in mitochondria, including mitochondrial DNA (mtDNA) maintenance, protein synthesis and degradation, assembly of the oxidative phosphorylation system, maintenance of cristae structure, and apoptosis[6]. Deletion of *Phb1* and *Phb2* did not cause any observable aberrant growth phenotype in *Saccharomyces cerevisiae*, but these proteins play a critical role in mitochondrial turnover and the lifespan of this yeast[7,8]. Prohibitins are required for embryonic development in *Caenorhabditis elegans*[9] and mouse[10]. Prohibitins also act as pleiotropic regulators of plant development. In *Arabidopsis thaliana*, Phb3 plays an essential role in ethylene signaling, nitric oxide accumulation, plant defense, induction of salicylic acid biosynthesis, cell division, senescence, and root meristem cell proliferation[11]. However, the roles of prohibitins in fungal pathogens are unknown.

Mitophagy is a cellular process responsible for selectively removing damaged or unwanted mitochondria[12]. The prohibitin Phb2 is essential for Parkin-mediated mitophagy by functioning as a mitochondrial receptor in mammalian cells, thereby affecting the recruitment of the autophagy protein LC3 to initiate mitochondrial clearance[13]. In yeast, the mitochondrial outer membrane protein Atg32 functions as a mitophagy-specific receptor that interacts with the core autophagic machinery Atg8 and Atg11 on the surfaces of mitochondria to induce mitophagy[14,15]. Although mitophagy is an evolutionarily conserved process, Atg32 homologs have only been identified in yeast. Therefore, it appears that ATG proteins that function in selective autophagic pathways, such as mitophagy, are not conserved like those in nonselective autophagic pathways in filamentous fungi[16]. In the filamentous fungus, *Magnaporthe oryzae*, mitophagy mediated by the sorting nexin MoAtg24 is essential for proper biotrophic development and invasive growth in rice (*Oryza sativa*)[17,18]. In *Fusarium graminearum*, FgSnx4 (also known as ATG24) is ubiquitously expressed at all developmental stages and localizes to early endosomes, and the FgSnx4 deletion mutant shows defects in polarized growth and virulence[19]. However, the regulatory mechanisms of mitophagy in fungal pathogenesis are not well understood.

*Colletotrichum higginsianum*, a hemibiotrophic fungal phytopathogen, is the causal agent of anthracnose disease in various cruciferous plants including the model plant *Arabidopsis thaliana*[20,21]. At the start of the hemibiotrophic lifecycle of *C. higginsianum* on *A. thaliana*, conidia contact the leaf surface and differentiate into melanized appressoria to facilitate penetration, followed by *in planta* growth, and formation of primary and secondary hyphae[22]. The *C. higginsianum–A. thaliana* pathosystem is an attractive model to examine hemibiotrophic fungal–plant interactions[23].

In the current study, we identified the prohibitin proteins ChPhb1 and ChPhb2 from *C. higginsianum* and demonstrated that these proteins can interact with each other in mitochondria and are involved in proper mitochondrial functioning, mitophagy and virulence. Moreover, we showed that the mitophagic protein, ChATG24, interacts with both ChPhb1 and ChPhb2 in the mitochondria and is also essential for mitophagy and virulence. Deletion of either *ChPhb1* or *ChPhb2* perturbs the translocation of ChATG24 into the mitochondria. Collectively, these findings

reveal that prohibitins recruit ChATG24 into the mitochondria to modulate mitophagy, thereby affecting fungal virulence.

## Results

**Characterization of ChPhb1 and ChPhb2.** To characterize prohibitin genes in a phytopathogenic fungus, we used the amino acid sequences of prohibitins (Phb1 and Phb2) from *S. cerevisiae* as queries for a BLASTP search against the *C. higginsianum* genome. Two proteins, Ch063_12292 and Ch063_08657, were identified as the *C. higginsianum* homologs of *S. cerevisiae* Phb1 and Phb2, respectively. Both ChPhb1 and ChPhb2 contain a conserved PHB domain (Fig. 1a).

Phylogenetic analysis of prohibitins from various organisms showed that ChPhb1 and ChPhb2 are divided into two distinct clades (Fig. 1b), suggesting that ChPhb1 and ChPhb2 are paralogous and that Phb1 and Phb2 are highly conserved in different species. We examined the expression profiles of *ChPhb1* and *ChPhb2* during each growth phase including vegetative hyphae, conidia, and the infection process on *A. thaliana* using qRT-PCR. *ChPhb1* and *ChPhb2* were highly expressed at the necrotrophic stage at 72 h post-inoculation (hpi) (Fig. 1c). These results suggested that prohibitin genes might play important roles in the formation of secondary hyphae during infection.

**ChPhb1 and ChPhb2 are required for full virulence.** To examine the functions of prohibitins in *C. higginsianum*, we generated both single and double deletion mutants of *ChPhb1* and *ChPhb2*, and their complemented strains (Supplementary Fig. 1). We analyzed the phenotypic changes in vegetative growth, conidiation, biomass, and appressorial formation in the deletion mutants, wild-type strain Ch-1 and complemented strains. The Δ*ChPhb2* deletion mutant displayed significantly attenuated growth and showed obviously reduced conidiation compared to the wild-type strain, while complementation of *ChPhb2* in Δ*ChPhb2* deletion mutant could restore growth and conidiation (Fig. 2a–d). However, the Δ*ChPhb2* deletion mutant showed no obvious defects in conidial germination or appressorial formation (Fig. 2e). No phenotypic changes were detected in the Δ*ChPhb1* deletion mutant (Fig. 2a–d). Moreover, the Δ*ChPhb1/ChPhb2* double deletion mutant showed significantly attenuated vegetative growth and conidiation, which was similar to the phenotype of the Δ*ChPhb2* deletion mutant (Fig. 2a–e). These results indicated that ChPhb2, but not ChPhb1, is involved in vegetative growth and conidiation.

To further explore the roles of ChPhb1 and ChPhb2 in host colonization and virulence, we conducted virulence and plant infection assays using the Δ*ChPhb1* and Δ*ChPhb2* single deletion mutants and the Δ*ChPhb1/ChPhb2* double deletion mutant on *A. thaliana* seedlings. At 7 dpi, typical water-soaked and collapsed lesions appeared on *Arabidopsis* leaves following inoculation with the wild-type and all complemented strains. In contrast, the Δ*ChPhb1*, Δ*ChPhb2*, and Δ*ChPhb1/ChPhb2* deletion mutants caused few and small necrotic lesions to form on infected leaves (Fig. 2f). We also assayed appressorial penetration and infectious growth in *Arabidopsis* leaves by microscopy. Compared to wild-type and complemented strains, the formation of primary hyphae was not significantly affected in Δ*ChPhb1* or Δ*ChPhb2*, but only 52% of the primary hyphae of Δ*ChPhb1* and 30% of the primary hyphae of Δ*ChPhb2* developed into secondary hyphae at 4 dpi (Fig. 2g, h). The Δ*ChPhb1/ChPhb2* double deletion mutant exhibited even less secondary hyphae formation than the single mutants, but there were no significant differences between Δ*ChPhb2* and Δ*ChPhb1/ChPhb2*. These results indicated that the *ChPhb1* and *ChPhb2* deletion mutants are defective in the

switch to necrotrophic growth and that *ChPhb2* might be more directly responsible for the reduced virulence.

**ChPhb1 interacts with ChPhb2 in mitochondria.** In budding yeast and mammals, Phb1 and Phb2 form large, multimeric ring complexes at the inner membranes of mitochondria[5,24]. In a yeast two-hybrid (Y2H) assay, blue colonies of Y2HGold with pGBKT7-ChPhb1 and pGAKT7-ChPhb2 were observed on SD–Leu–Trp–His/X-α-gal plates, demonstrating that ChPhb1 interacts with ChPhb2 (Fig. 3a). Previous studies in mammals showed that Phb1 and Phb2 can interact as a heterodimeric complex[8,13]. To further confirm the role of ChPhb1 and ChPhb2, yeast two-hybrid assays of ChPhb1 or ChPhb2 self-interaction were assessed and no self-interactions were found for ChPhb1 or ChPhb2 (Fig. 3a). These results indicated that ChPhb1 and ChPhb2 might form as a heterodimeric complex. This interaction between ChPhb1 and ChPhb2 was further validated by Co-IP assays in vivo. We generated ChPhb1–3×FLAG and ChPhb2-GFP fusion constructs and used them to co-transform the wild-type strain. Both ChPhb1–3×FLAG (46 kDa) and ChPhb2-GFP (61 kDa) fusion proteins were detected in the immunoprecipitate containing anti-FLAG M2 beads (Fig. 3b). The strain that only carried ChPhb1–3×FLAG was used as a control, and it was not detected in the immunoprecipitate (Fig. 3b). These results indicated that ChPhb1 interacts with ChPhb2 in vivo.

To examine the subcellular localizations of ChPhb1 and ChPhb2, we expressed *ChPhb1* and *ChPhb2* under the control of their native promoters fused with enhanced green fluorescent protein (eGFP) tags at their C termini and expressed them in the Δ*ChPhb1* and Δ*ChPhb2* mutants, respectively. The complementation strains which complement the phenotypes of deletion mutants were obtained (Fig. 2 and Supplementary Fig. 1). Punctate and tubular GFP fluorescent signals were observed in vegetative hyphae of the ChPhb1-GFP and ChPhb2-GFP tagged mutants. Upon staining with the red fluorescent mitochondrial dye MitoTracker, punctate or tubular GFP signals were seen to completely co-localized with the mitochondrial structures labeled by red fluorescence (Fig. 3c). These results indicated that both ChPhb1 and ChPhb2 localize to mitochondria.

To further enable direct visualization of interactions between ChPhb1 with ChPhb2, we performed bimolecular fluorescence complementation (BiFC) assays using ChPhb1 fused to the C-terminal fragment of YFP (YFP$^{CTF}$) and ChPhb2 fused to the N-terminal fragment of YFP (YFP$^{NTF}$). The two constructs expressing ChPhb1-YFP$^{CTF}$ and YFP$^{NTF}$-ChPhb2 were co-transformed into *C. higginsianum* strain Ch-1 while the constructs YFP$^{CTF}$ and YFP$^{NTF}$ were co-transformed into strain Ch-1 as a negative control. YFP signals were only detected in transformants co-expressing ChPhb1-YFP$^{CTF}$ and YFP$^{NTF}$-ChPhb2 but not in the negative control, suggesting that ChPhb1 interacts with ChPhb2 in mitochondria (Fig. 3d).

**ChPhb1 and ChPhb2 are important for mitochondrial structure and transmembrane potential.** Since ChPhb1 interacts with ChPhb2 in mitochondria, we explored whether the deletion of the two prohibitin proteins would result in defects in mitochondrial function. After staining with MitoTracker red, the mitochondria in hyphae showed elongated tubules in the wild-type and complemented mutant strains, whereas the red fluorescent signals were diffuse in the Δ*ChPhb1* and Δ*ChPhb2* deletion mutants as well as the Δ*ChPhb1/Phb2* double deletion mutant (Fig. 4a). We then expressed the mitochondria-localized fission protein ChFis1-GFP in the Δ*ChPhb1* and Δ*ChPhb2* deletion mutants to examine mitochondrial fission. Analysis of green fluorescent signals in mitochondria revealed elongated tubules in the hyphae, with no

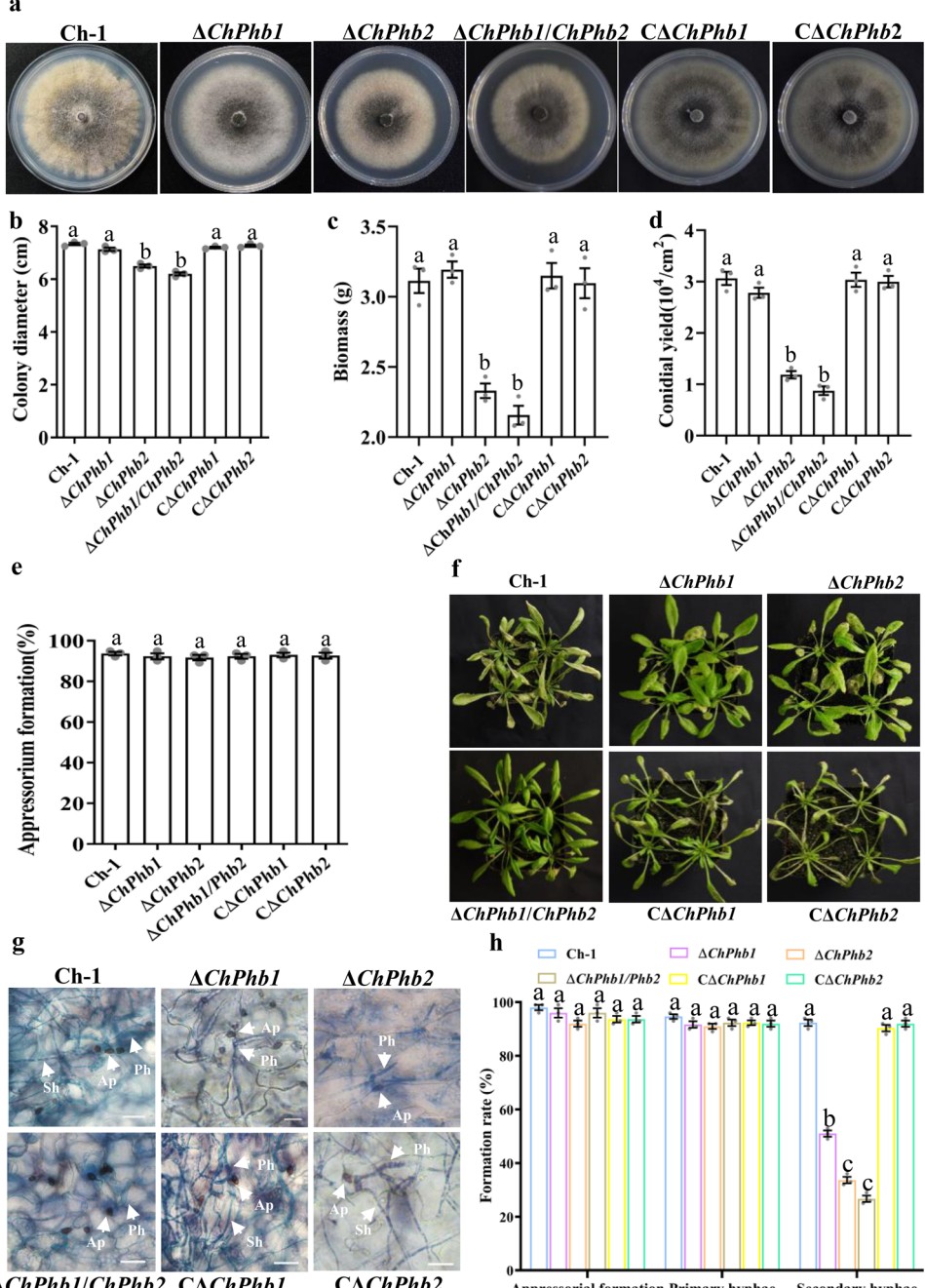

**Fig. 2 ChPhb1 and ChPhb2 are required for the full virulence of *C. higginsianum*. a** Morphology of colonies of each strain grown on PDA at 25 °C for 7 days. **b** Statistical analysis of colony growth. Colony diameter was measured after culturing at 25 °C for 7 days. **c** Statistical analysis of mycelial biomass. The biomass of each strain was measured after shake-cultured in PDB for 7 days. **d** Statistical analysis of conidial production. Conidiation was measured by collecting conidia from PDA plates after 7 days. **e** Appressorial formation rates of each strain on a hydrophobic surface. **f** Pathogenicity assay on *Arabidopsis* plants. Conidial suspensions of each strain were sprayed onto the plants and incubated at 25 °C for 4 days. **g** Infection structures of each mutant. Ap appressoria, Ph primary hyphae, Sh secondary hyphae. Scale bar, 20 μm. **h** Percentage of infection structure formation on Arabidopsis leaves. For bar graphs, error bars represent the standard deviation, and absence of lowercase letters in common indicate significant differences using LSD at $P = 0.05$.

difference detected among the strains (Supplementary Fig. 2). However, detailed transmission electron microscopy (TEM) analysis revealed defective morphogenesis of cristae in the absence of either prohibitin (Fig. 4b). These results suggest that ChPhb1 and ChPhb2 are crucial for maintaining mitochondrial structure but are not required for mitochondrial fission.

The intensity of the MitoTracker deep red changed along with mitochondrial transmembrane potential (ΔΨ)[25]. Thus, we tested whether mitochondrial ΔΨ was disturbed in the ΔChPhb1, ΔChPhb2, or ΔChPhb1/Phb2 deletion mutants using JC-1 dye. When treated with dye, an increase in red fluorescence was observed in the wild type, with a mean green/red fluorescence ratio of the probe reaching 3.57 ± 0.62, which is indicative of an increase in mitochondrial ΔΨ (Fig. 4c and Supplementary Fig. 3). Under similar exposure conditions, monomeric JC-1–associated green fluorescence in the ΔChPhb1, ΔChPhb2, and ΔChPhb1/ChPhb2

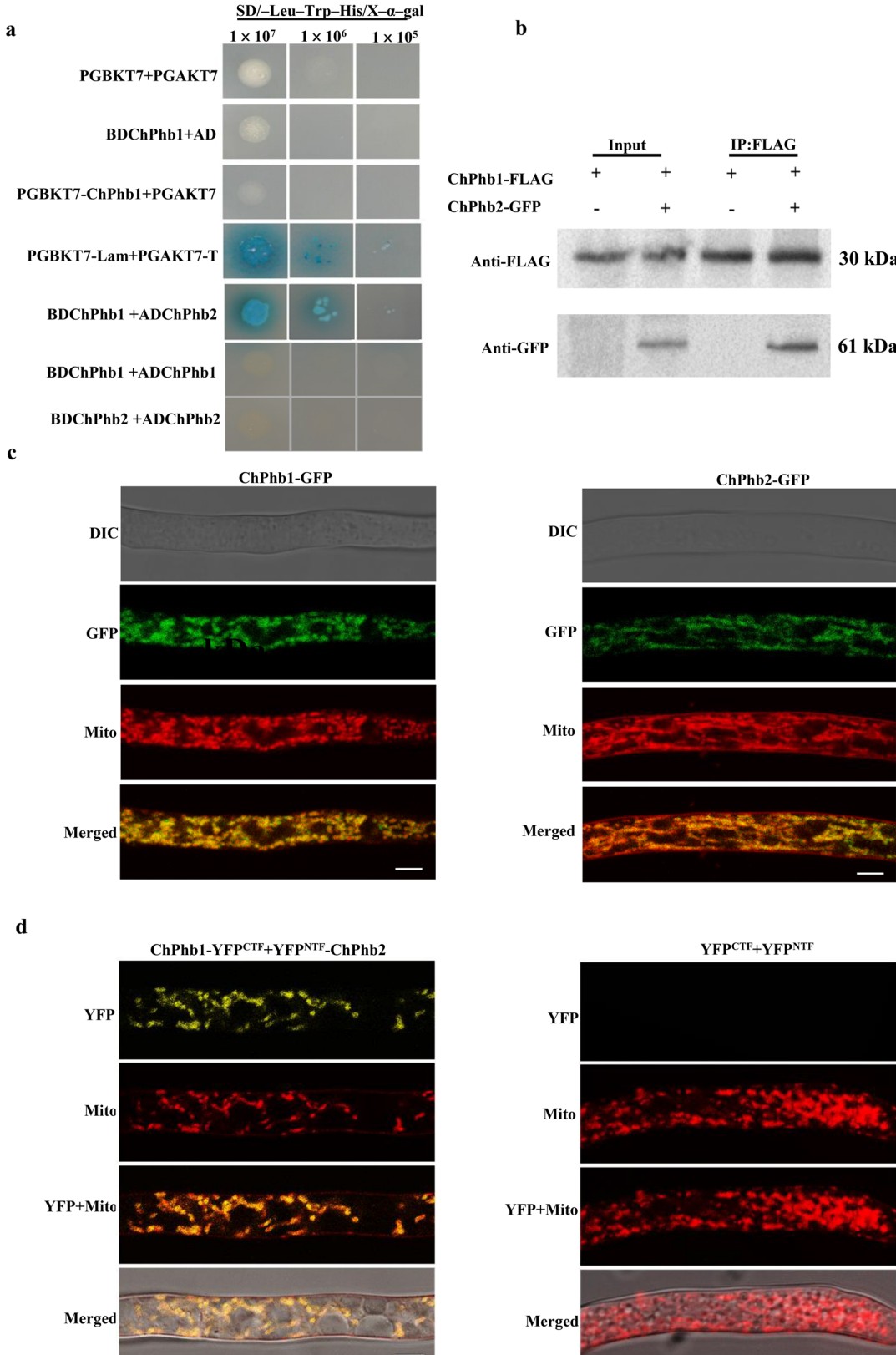

**Fig. 3 ChPhb1 interacts with ChPhb2 in mitochondria. a** The interaction of ChPhb1 and ChPhb2 was detected using an Y2H assay. Yeast transformants expressing plasmids harboring the indicated constructs were assayed for growth on nonselective plates (SD/-Leu/-Trp) and selective plates (SD/-Leu/-Trp/-His) with 1 mM X-gal for 3 days. **b** Co-IP assay of the interaction between ChPhb1 and ChPhb2. **c** Laser-scanning confocal microscope (LSCM) observation of the subcellular localizations of ChPhb1:GFP and ChPhb2:GFP in mitochondria of hyphae. DIC differential interference contrast, GFP green fluorescent protein. Scale bar, 5 μm. **d** BiFC assay of the interaction between ChPhb1 and ChPhb2. Hyphae were stained with Mito-tracker and then observed by confocal fluorescence microscope. Scale bar, 5 μm.

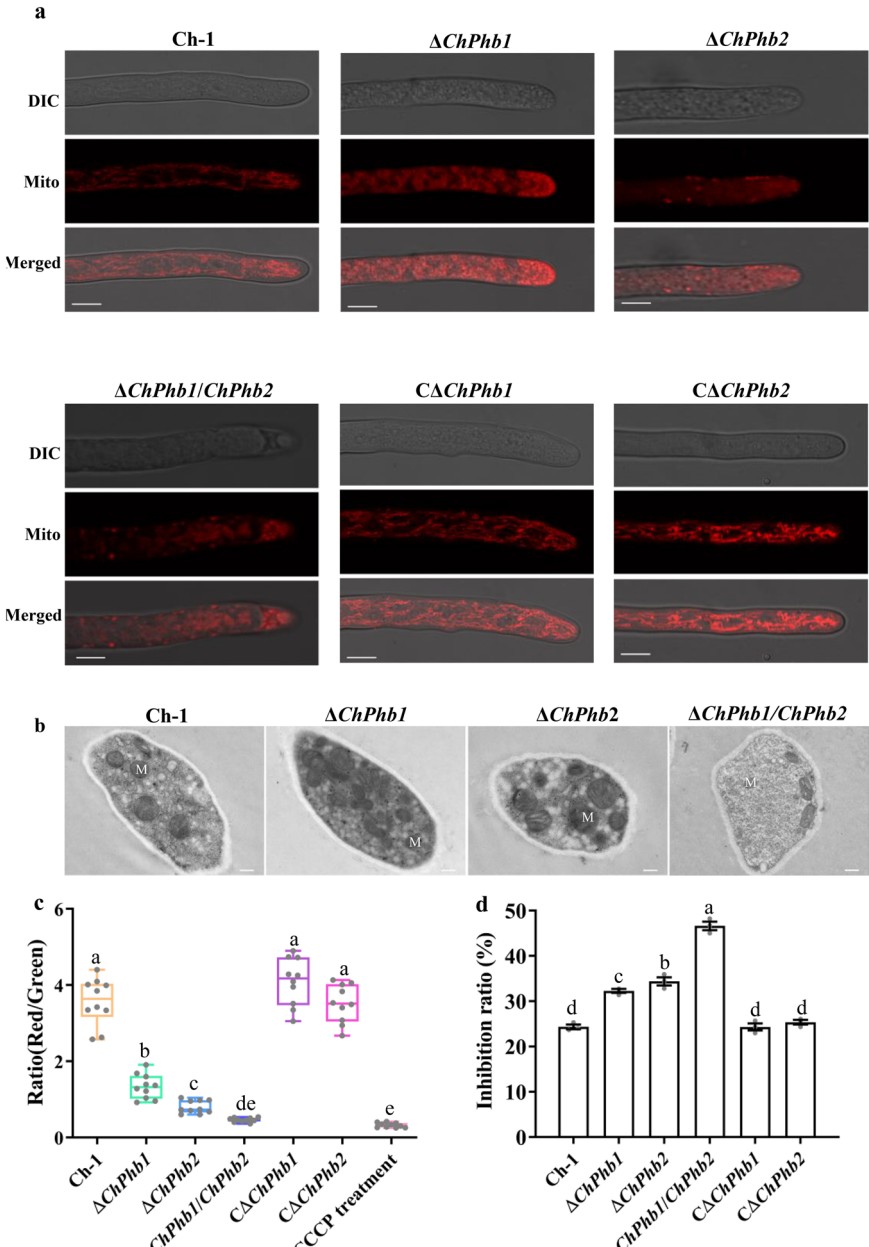

**Fig. 4 ChPhb1 and ChPhb2 are important for mitochondrial structure and function. a** Mitochondrial morphology in hyphal tips stained by Mito-tracker. Scale bar, 10 µm. **b** TEM observation of mitochondria in hyphae. Scale bar, 1 µm. **c** The mitochondrial membrane potential was evaluated by calculating the ratio of red to green fluorescence intensity for at least 10 fields under microscope. For each sample, box plot indicates median, lower and upper quartiles, and minimum and maximum values of ratio. For each box plot, the bottom and top horizontal lines indicate the first and third quartiles of the data, respectively. The horizontal line inside each box plot shows the median value. **d** Inhibition ratio calculated based on colony diameter of strains subjected to menadione stress for 7 days. Error bars represent the standard deviation. For bar graphs, error bars represent the standard deviation, and absence of lowercase letters in common indicate significant differences using LSD at $P = 0.05$.

deletion mutants was more pronounced compared to the wild type. Furthermore, the two single deletion mutants exhibited significantly higher green/red fluorescence ratio than the Δ*ChPhb1*/*ChPhb2* double deletion mutant. These results indicated that both ChPhb1 and ChPhb2 are involved in generating mitochondrial ΔΨ.

The loss of ΔΨ is thought to be directly associated with the accumulation of reactive oxygen species (ROS) [26]. To test this hypothesis, we examined the sensitivity of the Δ*ChPhb1* and Δ*ChPhb2* mutants to increased ROS levels. The Δ*ChPhb1*, Δ*ChPhb2*, and Δ*ChPhb1*/*ChPhb2* mutants showed increased sensitivity to menadione compared with the wild type (Fig. 4d and Supplementary Fig. 4). No significant changes were observed

between Δ*ChPhb1*, Δ*ChPhb2* or Δ*ChPhb1*/*ChPhb2* mutant and the wild type on the plates amended with $H_2O_2$ (Supplementary Fig. 4b). Menadione generates ROS specifically in mitochondria. These results indicated that ChPhb1 and ChPhb2 were required for responses to mitochondrial ROS, and overall suggested that these prohibitins are crucial for maintaining mitochondrial structure and function.

**ChPhb1 and ChPhb2 are involved in mitophagy.** Eukaryotes have evolved several quality control mechanisms that preserve mitochondrial homeostasis and prevent cellular damage. Mitophagy

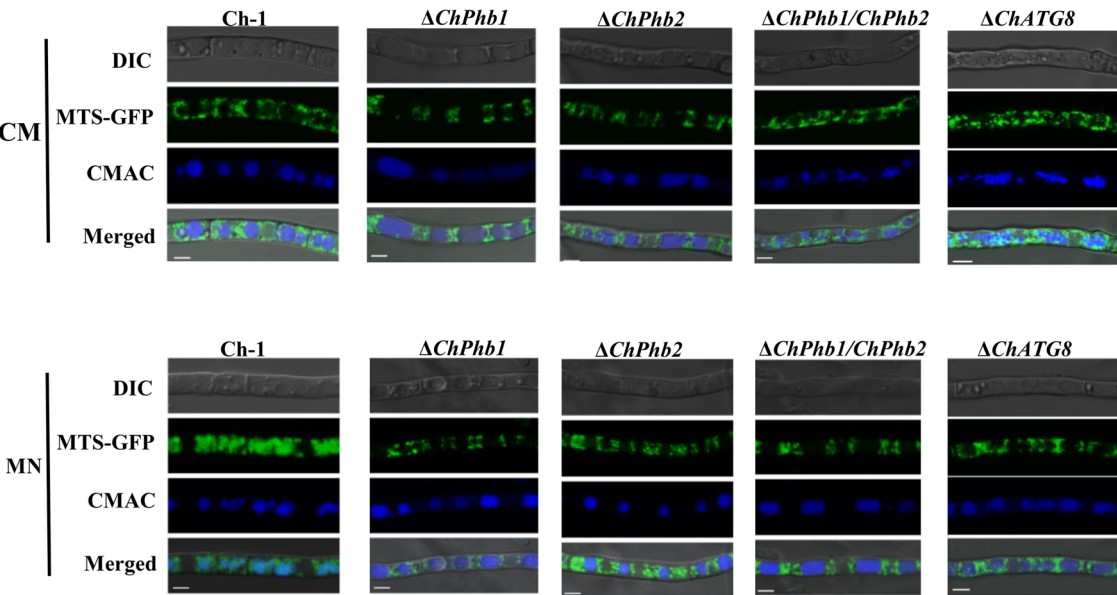

**Fig. 5 ChPhb1 and ChPhb2 are involved in mitophagy.** Mycelia from each strain were cultured in CM for 2 days, shifted to BM-G for 30 h and then starved in MM-N for 12 h. The strains subjected to MN starvation were co-stained with 10 µM CMAC. Confocal microscopy with Mito-GFP and CMAC-stained vacuoles was performed to detect mitophagy in each strain. Scale bar, 5 µm.

is a cellular process that selectively removes mitochondria through autophagy; this process is triggered by the presence of severely damaged or superfluous mitochondria[27]. We generated Δ*ChPhb1* and Δ*ChPhb2* deletion mutants carrying Mito-GFP and used them to monitor mitophagy. The two Mito-GFP strains were cultured in liquid CM for 2 days, then shifted to BG for 48 h to induce mitochondrial biogenesis, and shifted to MN for 12 h to induce mitophagy. In glycerol medium, GFP signals were randomly distributed in the cytoplasm of the mycelia, except for the vacuoles. When the mycelia were transferred from glycerol medium to nitrogen-starvation medium, GFP signals were clearly observed in mycelial vacuoles of the wild type but not of the Δ*ChPhb1*, Δ*ChPhb2* or Δ*ChPhb1/ChPhb2* mutants (Fig. 5). These results indicated that mitophagy was blocked upon the loss of ChPhb1 or ChPhb2.

To determine whether ChPhb1 and ChPhb2 are involved in non-specific autophagy, we expressed GFP-ATG8 in the wild type and deletion mutants to monitor non-specific autophagy. In the Δ*ChPhb1* and Δ*ChPhb2* deletion mutants, a vacuolar GFP signal was detected after starvation in MN medium for 5 h, just as with the wild-type strain (Supplementary Fig. 5), suggesting that ChPhb1 and ChPhb2 are not essential for non-specific autophagy. Since previous studies reported that ATG8 is essential for all types of autophagy, we generated a Δ*ChATG8* deletion mutant carrying Mito-GFP. The results showed that no vacuolar GFP signal was observed in the Δ*ChATG8* mutant (Fig. 5), suggesting that delivery of Mito-GFP in *C. higginsianum* to the vacuole indeed required *ChATG8*. Overall, these results suggested that ChPhb1 and ChPhb2 are specifically involved in mitophagy in *C. higginsianum*.

**ChATG24 interacts with ChPhb1 and ChPhb2 in the mitochondria.** To further explore how ChPhb1 and ChPhb2 participate in mitophagy, we performed Y2H assays to examine their interactions with the ATG proteins, ATG7, ATG8, ATG11, and ATG24, which might be associated with mitophagy in pathogenic fungi[17,28,29]. We co-transformed yeast cells with various combinations of AD and BD constructs and selected the cells on SD–Leu–Trp–His. Only ChATG24 was found to interact with ChPhb1 and ChPhb2 (Fig. 6a).

To validate the interactions between ChATG24 and prohibitin proteins in vivo, we performed Co-IP assays. We generated the ChPhb1–3×FLAG and ChATG24-GFP fusion constructs and introduced them into wild-type strain Ch-1 via co-transformation. Immunoblot analysis with anti-FLAG and anti-GFP antibodies detected the presence of a 30-kDA or a 73-kDa band, respectively. In proteins eluted from anti-FLAG M2 beads, the 73-kDa ChATG24-GFP band was detected in the transformant expressing the ChPhb1–3×FLAG and the ChATG24-GFP constructs, but not in the transformant expressing the ChPhb1–3×FLAG construct only (Fig. 6b). Similar methods were used to test the transformant expressing ChATG24-3×FLAG with ChPhb2-GFP fusion constructs (Fig. 6c). These results suggest that both ChPhb1 and ChPhb2 interact with ChATG24 in vivo.

To further confirm the interactions between ChPhb1 and ChPhb2, ChPhb1 and ChAtg24, or ChPhb2 and ChAtg24 in the absence of *ChATG24*, *ChPhb2* or *ChPhb1*, three fusion construct sets of *ChPhb1–3×FLAG* and *ChPhb1-GFP*, *ChPhb1–3×FLAG* and *ChATG24-GFP*, or *ChATG24-3×FLAG* and *ChPhb2-GFP* were separately transformed into the Δ*ChATG24*, Δ*ChPhb2* or Δ*ChPhb1* mutant. Results of Co-IP assays showed that ChPhb1 interacted with ChPhb2 in the Δ*ChATG24* mutant, ChPhb1 interacted with ChAtg24 in the Δ*ChPhb2* mutant, and ChPhb2 interacted with ChAtg24 in the Δ*ChPhb1* mutant (Fig. 6d), suggesting that the absence of any one of these three proteins did not affect the interactions between the other two.

We further confirmed the interactions between ChATG24 and the two prohibitin proteins using BiFC assays. Constructs expressing ChPhb1-YFP[CTF] and YFP[NTF]-ChATG24 were co-transformed into wild-type *C. higginsianum* to test the ChATG24-ChPhb1 interaction, and constructs expressing ChATG24-YFP[CTF] and YFP[NTF]-ChPhb2 were co-transformed into wild-type *C. higginsianum* to test the ChATG24-ChPhb2 interaction. YFP signals were only detected in transformants co-expressing ChPhb1-YFP[CTF] and YFP[NTF]-ChATG24 or ChATG24-YFP[CTF] and YFP[NTF]-ChPhb2, but not in the negative control (Fig. 6d), suggesting that ChATG24 interacts with both ChPhb1 and ChPhb2. After staining with the mitochondrial dye MitoTracker, YFP fluorescence was observed as punctate or tubular patterns that completely co-localized with the mitochondrial structures showing red fluorescence. These results indicated

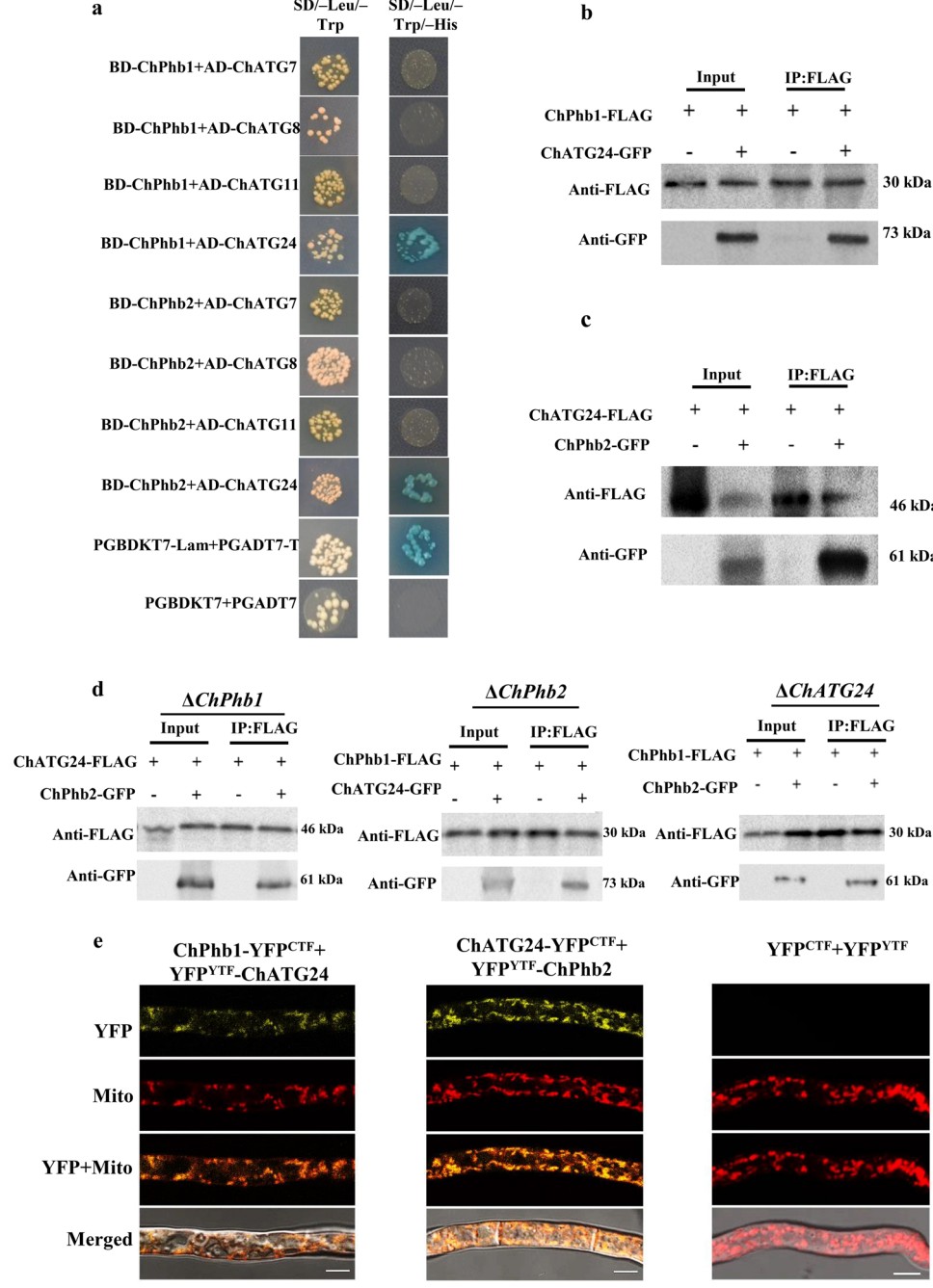

**Fig. 6 ChATG24 interacts with ChPhb1 and ChPhb2 in mitochondria. a** Interactions between ChATG24 and ChPhb1/ChPhb2 detected using an Y2H assay. Yeast transformants expressing plasmids harboring the indicated constructs were assayed for growth on selective plates (SD/−Leu/−Trp/−His) and nonselective plates (SD/−Leu/−Trp) and for β-galactosidase (LacZ) activity. **b** Co-IP assays of the interactions between ChATG24 and ChPhb1. Asterisks represent the tagged bands. **c** Co-IP assays of the interactions between ChATG24 and ChPhb2. Asterisks represent the tagged bands. **d** Co-IP assay in ChPhb1, ChPhb2, or ChAtg24 single deletion mutant to confirm the interactions between the other two of the three proteins (ChPhb1, ChPhb2, and ChAtg24) in the absence of one of them. **e** Visualization of the interaction between ChATG24 and ChPhb1/ChPhb2 in BiFC assay. Vegetative hyphae were stained with Mito-tracker and then analyzed by confocal fluorescence microscope. Scale bar, 5 μm.

that ChATG24 interacts with the two prohibitin proteins, ChPhb1 and ChPhb2, in mitochondria.

**ChATG24 is also important for fungal virulence and mitophagy.** Since the role of ChATG24 in *C. higginsianum* has not been fully elucidated, we further characterized it using a targeted gene replacement strategy with the ΔChATG24 deletion mutant (Supplementary Fig. 6). Fungal growth and conidiation were significantly reduced in this mutant compared to the wild type

(Fig. 7a–c). However, compared to the wild type, the ΔChATG24 deletion mutant showed no obvious defects in conidial germination or appressorial formation. In pathogenicity assays, the ΔChATG24 deletion mutant showed reduced virulence on *A. thaliana* compared to the wild type (Fig. 7d). In addition, the ΔChATG24 deletion mutant was defective in the switch to necrotrophic growth (Fig. 7e–f).

To determine whether ChATG24 is required for mitophagy, we generated ΔChATG24 deletion mutants carrying Mito-GFP. In

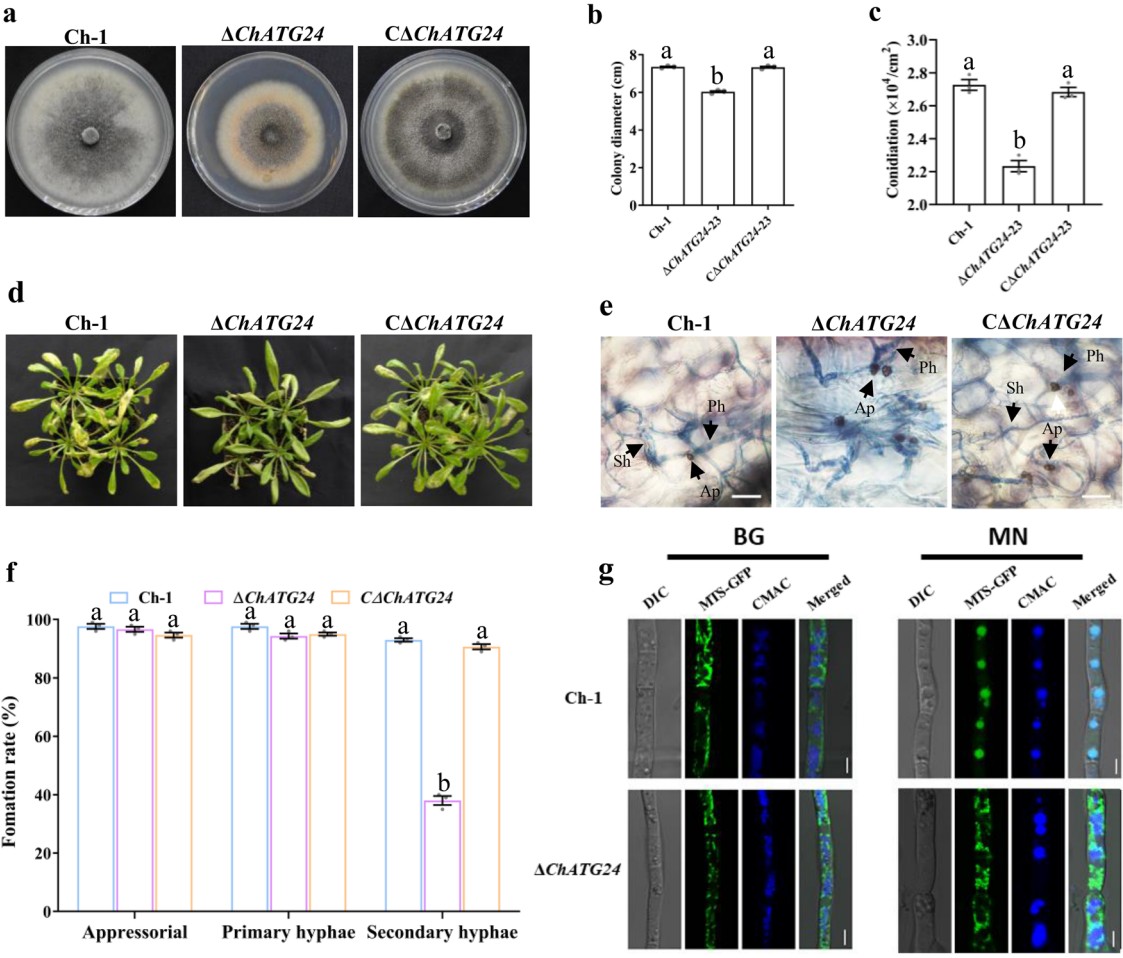

**Fig. 7 ChATG24 is important for fungal development and virulence. a** Morphology of colonies of each strain grown on PDA for 7 days. **b** Statistical analysis of colony growth. Diameter was measured after culturing at 25 °C for 7 days. **c** Statistical analysis of conidial production. Conidiation was measured by collecting conidia from PDA plates after 7 days. **d** Pathogenicity assays of each strain by spraying conidial suspensions onto Arabidopsis plants. The inoculated plants were incubated at 25 °C for 4 days. **e** Infection structures observed under microscope. Ap appressoria, Ph primary hyphae, Sh secondary hyphae. Scale bar, 20 μm. **f** Percentage of infection structure formation on *Arabidopsis* leaves. **g** ChATG24 is involved in mitophagy. Mycelia from each strain subjected to BG/MN medium were co-stained with 10 μM CMAC. Confocal microscopy of Mito-GFP and CMAC-stained vacuoles was performed to detect mitophagy in each strain. Scale bar, 5 μm. For bar graphs, error bars represent the standard deviation, and absence of lowercase letters in common indicate significant differences using LSD at $P = 0.05$.

mitophagy monitoring assays, GFP signals were randomly distributed in the cytoplasm (except for vacuoles) of mycelia in glycerol medium. Following transfer to MN medium, green fluorescent signals were observed in mycelial vacuoles of the wild type but not the Δ*ChATG24* deletion mutant (Fig. 7g), indicating that mitophagy was defective in the mutant. These results indicated that ChATG24 is essential for hyphal growth, conidiation, virulence, and mitophagy.

**Deletion of *ChPhb1* or *ChPhb2* perturbs the translocation of ChATG24 into mitochondria.** To examine the subcellular localization of ChATG24, we generated a *C. higginsianum* strain expressing *ChATG24-GFP* under the control of its native promoter. ChATG24-GFP was undetectable in mycelia cultured in liquid CM. In a previous study, ATG24 was not expressed in CM, but it was expressed and localized to mitochondria under nitrogen starvation conditions [17]. Therefore, we cultured the Δ*ChATG24* deletion mutant introduced harboring *ChATG24-GFP* under nitrogen starvation conditions and examined the subcellular localization of the fusion protein. Upon induction by nitrogen starvation, the ChATG24-GFP signal was diffusely distributed in the cytoplasm where it aggregated as punctate

structures. After labeling mitochondria with MitoTracker, ChATG24-GFP partially co-localized with MitoTracker-stained mitochondria (Fig. 8a). After labeling lipid droplets with Nile Red, ChATG24-GFP co-localized with the stained lipid droplets (Fig. 8a). Overall, these results indicate that ChATG24 primarily localizes to the cytoplasm in lipid droplets and mitochondria under nitrogen starvation conditions.

To further confirm the notion that prohibitin is a mitophagy cargo of ChATG24, we examined the subcellular localization of GFP-ChATG24 in the Δ*ChPhb1* and Δ*ChPhb2* deletion mutants. The ChATG24-GFP signal in both the Δ*ChPhb1* and Δ*ChPhb2* deletion mutants was diffusely distributed in the cytoplasm and aggregated as punctate structures, but the punctate structures in the mutants were significantly larger than those of the wild type (Fig. 8b). A co-localization assay was carried out with ChATG24-GFP and Mito-tracker. Fluorescence observation and fluorescence intensity analysis showed that most ChATG24-GFP appeared in mitochondria in Ch-1 while only a small part of ChATG24-GFP appeared in mitochondria in *ChPhb1* or *ChPhb2* deletion mutants (Fig. 8c). These results suggested that the deletion of *ChPhb1* or *ChPhb2* perturbs the translocation of ChATG24 into the mitochondria.

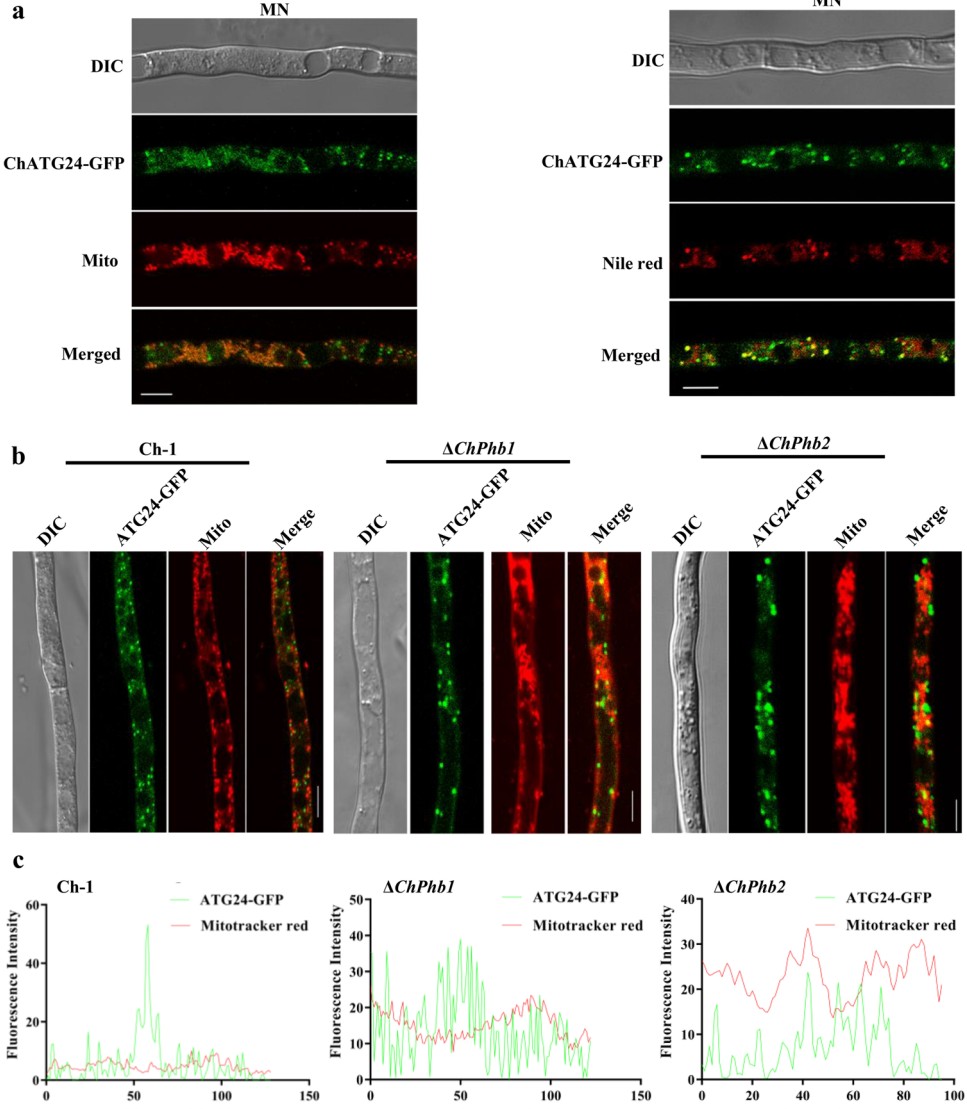

**Fig. 8 Deletion of *ChPhb1* or *ChPhb2* perturbs the translocation of ChATG24 into mitochondria. a** LSCM observation of the subcellular localization of ChATG24:GFP under nitrogen-starvation conditions. **b** LSCM observation of the subcellular localization of ChATG24:GFP in wild type Ch-1 and in *ChPhb1* and *ChPhb2* deletion mutants. DIC differential interference contrast, GFP green fluorescent protein. Scale bar, 5 μm. **c** Linescan graph consistent with the mitochondria localization of ChATG24 in the Δ*ChPhb1* and Δ*ChPhb2* mutant under starvation condition (MN).

**ChATG24 rescues the defects of the mitophagy-deficient mutant Δ*MoATG24*.** MoATG24 is a mitophagy-specific receptor that localizes to the mitochondria and is required for virulence in *Magnaporthe oryzae*[17]. Mitophagy mediated by MoATG24 is required for the proper invasive growth of *M. oryzae*[18]. Thus, to examine whether the function of ChATG24 in mitophagy is conserved in plant pathogenic fungi, we expressed *ChATG24* under the control of the constitutive promoter RP27 in Δ*MoATG24*. The growth defect was completely rescued in Δ*MoATG24* expressing ChATG24 (Fig. 9a, b).

To investigate whether *ChATG24* could rescue the pathogenicity of Δ*MoATG24*, we inoculated susceptible rice seedlings (*Oryza sativa* cv. LTH) with the complementation mutant at the fifth leaf stage by spraying with a conidial suspension. Compared to the Δ*MoATG24* mutant, which only occasionally produced small lesions, the complementation strains com *ChATG24-2* and com *ChATG24-11* gave numerous typical necrotic lesions (Fig. 9c).

To examine the differences between plant infection by Δ*MoATG24* and the complementation strains, we observed invasive hyphae under the microscope at 24 and 48 hpi (Fig. 9d, e). At 24 hpi, less than 1% of Δ*MoATG24* appressoria were capable of invading the rice sheath. In contrast, nearly 76% of appressoria of the complementation strains successfully penetrated the rice sheath. At 48 hpi, the penetration rates of appressoria of the wild type (P131) and complemented mutants com *ChATG24-2* and com *ChATG24-11* were comparable: 71% of wild-type invasive hyphae and 67% and 66% of invasive hyphae in com *ChATG24-2* and com *ChATG24-11*, respectively, formed branches. By contrast, only 1% of invasive hyphae in Δ*MoATG24* formed branches. Thus, ChATG24 restored the pathogenicity of Δ*MoATG24*, suggesting that ChATG24 shares overlapping functions with MoATG24.

## Discussion

Prohibitins are eukaryotic proteins with highly conserved functions, playing key roles in transcriptional regulation during cell proliferation, differentiation, and cellular aging; regulation of sister chromatid cohesion; cellular signaling apoptosis; and mitochondrial biogenesis[8]. The prohibitin complex has a variety

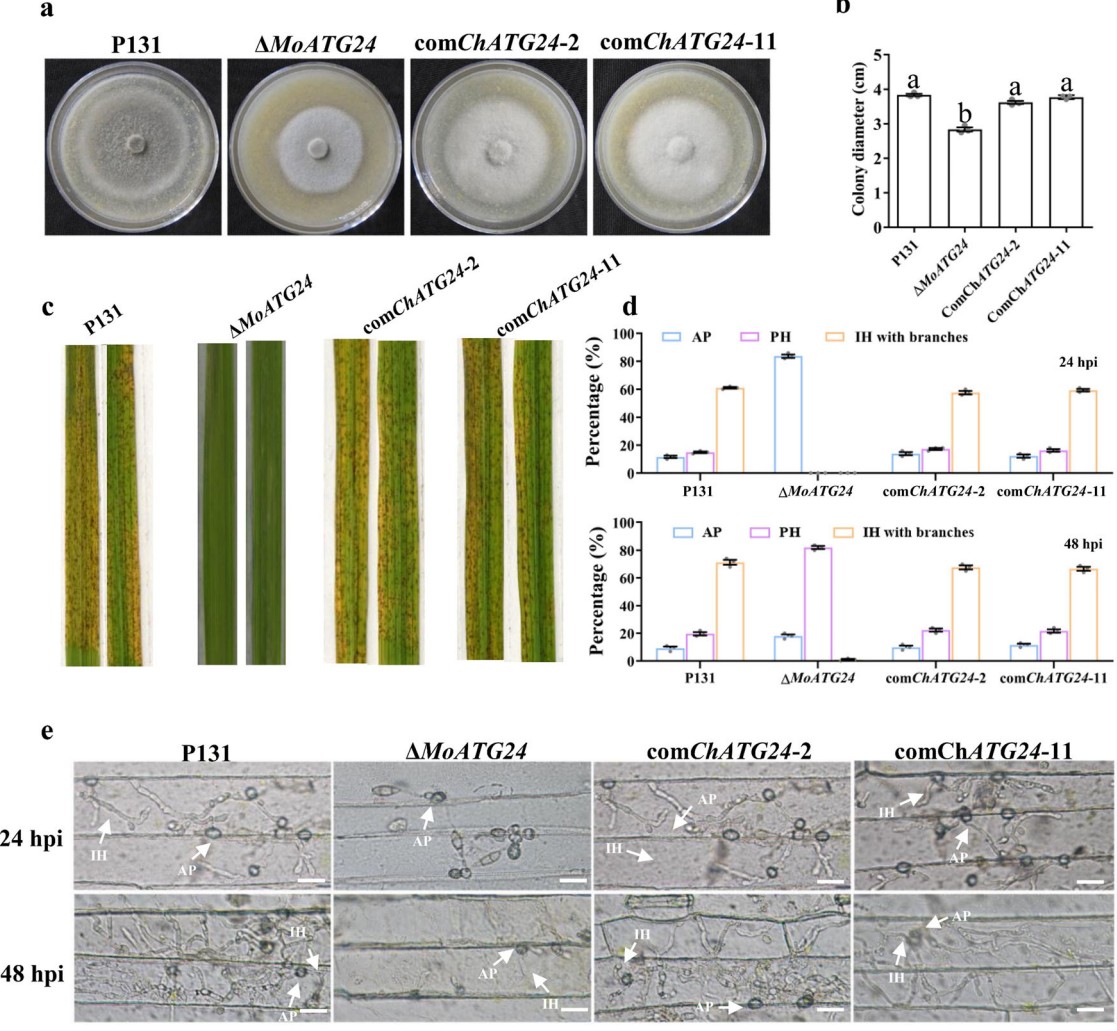

**Fig. 9 ChATG24-GFP restored virulence in ΔMoATG24. a** Colony morphology of the wild-type P131, ΔMoATG24 mutant and complemented transformant on oatmeal tomato agar. **b** Colony diameter of each strain measured after 7 days at 25 °C. **c** Lesions formed on rice leaves 5 dpi after inoculation with the indicated strains. **d** Formation rate of different infection structures in rice epidermal cells at 24 and 48 hpi. Percentages of appressoria (AP), primary infectionhyphae (PH), invasive hyphae (IH) with one to three branches and more than three branches formed by different strains were calculated. **e** Observation of the invasive hyphal growth of the strains on rice leaves at 24 and 48 hpi. IH of different strains formed in rice epidermal cells were observed at 24 and 48 hpi. Scale bar, 20 μm. For bar graphs, error bars represent the standard deviation, and absence of lowercase letters in common indicate significant differences using LSD at $P = 0.05$.

of putative functions within mitochondria. In this study, we characterized the prohibitin genes *ChPhb1* and *ChPhb2* in *C. higginsianum* and found a pathogenic mechanism linking prohibitin and mitophagy in this fungal pathogen.

Phb1 and Phb2 share more than 50% identical amino acid residues. Conserved domains shared within a gene family usually exhibit common functions. In the present study, we found that *C. higginsianum* Phb1 and Phb2 both contain a PHB domain, which is characteristic of the SPFH family of membrane proteins. Phb2, a subunit of the prohibitin complex, exerts many relevant functions via interactions with Phb1[30]. However, our results suggested that ChPhb2, but not ChPhb1, is involved in vegetative growth and conidiation. The prohibitin complex is anchored to the mitochondrial inner membrane via its N-terminal hydrophobic regions[31]. The N-terminal hydrophobic regions of ChPhb1 and ChPhb2 differ, perhaps explaining why ChPhb2 showed some distinct functions from ChPhb1 in *C. higginsianum*.

In this study, we found that there were no significant differences between Δ*ChPhb2* and Δ*ChPhb1*/*ChPhb2* mutants in vegetative growth, conidiation or pathogenicity. However, the

Δ*ChPhb1*/*ChPhb2* deletion mutant showed significant damage in mitochondrial transmembrane potential and reduced menadione sensitivity compared to Δ*ChPhb2* mutant. The degree of mitochondrial damage caused by deletion of *ChPhb1*, *ChPhb2*, or *ChPhb1*/*ChPhb2* did not correspond to their respective vegetative growth or conidiation. These indicated that *ChPhb1* and *ChPhb2* functioned not only individually but also collaboratively in mitochondria. Previous studies showed that prohibitin1 and prohibitin2 function as a heterodimeric complex[5]. Our Y2H results indicated that ChPhb1 and ChPhb2 function presumably as a heterodimeric complex in *C. higginsianum*, which was consistent with the findings of most studies in human tissues and yeasts[8,13]. This could explain why ChPhb1 and ChPhb2 function independently.

Phylogenetic analysis indicated that Phb1 and Phb2 in *C. higginsianum* share high homology with prohibitins of human, mouse, and other fungi, suggesting that the regulatory functions of prohibitins in various species might also be highly similar. However, defects in prohibitins do not cause any observable growth phenotypes in *S. cerevisiae*[32]. Prohibitins are required for

embryonic development in *C. elegans* and mouse[9,10]. In contrast, ChPhb2 is required for vegetative growth and conidiation in *C. higginsianum*, indicating that the roles of prohibitin in cellular differentiation may differ markedly among animals, yeasts, and phytopathogenic fungi.

Prohibitins are highly conserved mitochondrial proteins that have a variety of putative functions within mitochondria. The prohibitin complex has been associated with mtDNA maintenance, mitochondrial biogenesis and degradation, biogenesis and assembly of the oxidative phosphorylation system, maintenance of the cristae structure, and membrane lipid homeostasis[8]. In the current study, ChPhb1 and ChPhb2 were shown to localize to the mitochondria. This is in line with the finding that Phb1 and Phb2 localize to the mitochondrial inner membranes in the cells of various eukaryotic organisms[6,33,34].

We provided several lines of evidence that deletion of ChPhb1 or ChPhb2 causes mitochondrial dysfunction, which results in increased ROS synthesis. First, ChPhb1 and ChPhb2 were verified to be located in the mitochondria; however, in a MitoTracker staining experiment, diffuse red fluorescent signals in the cytoplasm were detected in the Δ*ChPhb1* and Δ*ChPhb2* mutants. In a previous study, upon the depletion of prohibitin, mitochondria appeared fragmented and disorganized[10]. These findings suggested that the deletion of ChPhb1 and ChPhb2 causes mitochondrial dysfunction. Second, a detailed TEM analysis revealed defective morphogenesis of cristae in the absence of either prohibitin in *C. higginsianum*. Similarly, defective morphogenesis of cristae has been reported in plant and yeast mitochondria depleted of prohibitin[10,35]. Third, the mitochondrial $\Delta\Psi$ was disordered in the Δ*ChPhb1* and Δ*ChPhb2* mutants. This observation is consistent with the finding that in yeast and human cells, the loss of the prohibitin complex results in a defect in mitochondrial $\Delta\Psi$[36–38]. We attempted many times to measure mitrochondrial ROS levels levels in these strains using DCFH-DA and mitoSOX-red. However, we were unable to obtain consistent data either when attempting to directly detect in situ or when using isolated mitochondria of *C. higginsianum*. Future work is needed to develop a more efficient method of ROS detection in mitochondria of *C. higginsianum*.

The loss of $\Delta\Psi$ is directly associated with the accumulation of ROS[8]. Exogenously added menadione specifically generates ROS in mitochondria[39,40]. The *ChPhb1* and *ChPhb2* deletion mutants showed strongly increased sensitivity to menadione compared to the wild type. These results suggest that the deletion of *ChPhb1* and *ChPhb2* significantly increased ROS levels in mitochondria. This is in line with the previous finding that a lack of prohibitin results in increased ROS production[9]. Thus, the role of prohibitin in maintaining proper mitochondrial function is highly conserved in eukaryotes.

The disruption of mitophagy results in distorted mitochondrial function, mitochondrial membrane depolarization, decreased ATP levels, increased oxygen utilization, and elevated ROS levels[41]. Phb2 is involved in mitochondrial homeostasis[10] and plays a critical role in mitophagic responses in evolutionarily distant models, such as human cancer cells and *C. elegans*[13]. The current study provided evidence that mitophagy was delayed in Δ*ChPhb1* and Δ*ChPhb2*. Cell biology analyses proved that mitophagy is significantly induced under nitrogen starvation condition. Mitophagy is quite different from nonselective macroautophagy, which typically engulfs cellular components including mitochondria[42,43]. This is consistent with the finding that Phb2 is a crucial mitophagy receptor involved in targeting mitochondria for autophagic degradation in mammalian cells[13]. Furthermore, exogenously added menadione specifically induces ROS formation in mitochondria[39,40]. In the current study, *ChPhb1* and *ChPhb2* deletion mutants showed conspicuously

increased sensitivity to mitochondrial ROS, but not to exogenous $H_2O_2$. This observation confirms the roles of ChPhb1 and ChPhb2 in mitophagy.

Mitophagy plays essential roles in cellular differentiation and pathogenesis in filamentous fungi. For example, mitophagy is essential for conidial differentiation and fungal pathogenesis in *M. oryzae*[17,18]. Similarly, in *A. oryzae*, the mitophagy-related protein ATG26 is involved in conidiation, growth of aerial hyphae, and invasive growth[44]. Otherwise, mitophagy is also required for the development of blastopores and starvation stress in *Beauveria bassiana*[45]. However, the relationship between mitophagy and fungal pathogenesis remains elusive and needs to be further explored.

The conserved nature of ATG24 proteins in yeast and other filamentous fungi demonstrates a deep evolutionary heritage in eukaryotes[19]. In *M. oryzae*, MoATG24-GFP co-localizes with mitochondria under nitrogen starvation conditions[17]. In *F. graminearum*, FgATG24-GFP is ubiquitously expressed during all stages of development and localizes to early endosomes[19]. Nevertheless, in the current study, ChATG24-GFP localized to the cytoplasm with lipid bodies under nitrogen starvation conditions. This difference in the intracellular localization of ATG24 among *C. higginsianum*, *M. oryzae*, and *F. graminearum* could account for its specialized function in each organism. The fungal growth defect and virulence caused by deletion of ATG24 appears to be consistent among *C. higginsianum*, *M. oryzae*, and *F. graminearum*. MoATG24-mediated mitophagy is required for the proper invasive growth of *M. oryzae*[18]. To explore the conservation of ATG24 in mitophagy, we expressed GFP-ChATG24 under the control of the strong constitutive promoter RP27 in the Δ*MoATG24* mutant. The growth defects and virulence were completely rescued in Δ*Moatg*24 expressing ChATG24-GFP, indicating that ATG24 plays a relatively conserved role in mediating mitophagy in *C. higginsianum* and *M. oryzae*.

During selective autophagy, the recognition of a specific cargo depends on a series of receptor proteins (e.g., ATG19p in cytoplasm-to-vacuole targeting and ATG32p in mitophagy). Such receptors sense stimuli that induce mitophagy and couple mitochondrial dynamics to the quality control machinery[46]. In the current study, we uncovered a direct relationship between prohibitin and ChATG24. Co-IP and BiFC assays provided evidence that both ChPhb1 and ChPhb2 interact with ChATG24 in the mitochondria. Moreover, Δ*ChATG24* is defective in mitophagy. Our findings suggest that prohibitin regulate fungal virulence by mediating ChATG24-assisted mitophagy.

This study provides insight into the relationship between prohibitin and mitophagy in a fungal pathogen, and reveals a new mitophagy regulation mechanism: prohibitin acts as a ChATG24-assisted mitophagy regulator. To the best of our knowledge, this is the first functional analysis of prohibitins in a phytopathogenic fungus. These findings expand our understanding of the mitophagy mechanism in fungi.

## Methods

**Fungal strains, media and growth conditions.** *Colletotrichum higginsianum* strain IMI349061 (Ch-1) is fully virulent on *A. thaliana* ecotype Col-0 and was used as the reference wild-type strain in all experiments. All strains derived from the reference strain were maintained and cultured on potato dextrose agar (PDA) at 25 °C in the dark. All *C. higginsianum* strains used in this study are listed in Supplementary Data 2. For fungal DNA or RNA isolation, five-day-old mycelia collected from liquid cultures grown in potato dextrose broth (PDB) under shaken conditions (180 rpm) were used. Conidia washed from the culture were used for *A. tumefaciens*-mediated transformation (ATMT). To generate the gene deletion and complementation mutants, the transformants were isolated and screened on PDA supplemented with 50 μg/ml hygromycin B (Roche, Mannheim, Germany) or 150 μg/ml neomycin (Ameresco, OH, USA), respectively.

**Deletion and complementation of target genes.** To replace the *ChPhb1* gene, ~1.1 kb of flanking sequences of upstream and downstream of *ChPhb1* were amplified with specific primers (Supplementary Data 2). The PCR product of the upstream and downstream flanking sequence were ligated into pMD18T-HYG vector to generate the initial vector pMD18T-F1-HYG-F2. This vector was digested with HindIII and KpnI and ligated with pNeo3300III. The resulting constructs were transformed into the wild-type strain Ch-1 via ATMT[47]. The hygromycin resistant transformants were further confirmed by reverse-transcription PCR (RT-PCR) analysis (Supplementary Data 2) and Southern blotting according to the manufacturer's instructions (Amersham Gene Images Alkphos Direct Labelling and Detection System, GE Healthcare, UK). *ChPhb2* and *ChATG24* deletion mutants were generated using the same strategy. To generate *ChPhb1/ChPhb2* double knockout mutants, the F1-HYG-F2 sequence of *ChPhb2* was ligated with the pNeo3300III-bar vector using a similar strategy. The resulting constructs were transformed into the *ΔChPhb1* mutant. The putative knockout mutants with hygromycin B resistance and bialaphos resistance (bar) were further confirmed by RT-PCR analysis (Supplementary Data 2).

To construct the complementation vector, the *ChPhb1* complement fragment, containing its native promoter region and the entire gene coding region without the termination codon was cloned into pNeo3300-GFP vector. The complementation vector was transformed using ATMT by co-culture with the *ΔChPhb1* deletion mutant. Putative complementation mutants with G418 resistance were selected for RT-PCR (Supplementary Fig. 1). Similar methods were used to complement the *ΔChPhb2* and *ΔChATG24* deletion mutants.

**Phenotypic analyses.** Mycelial plugs (9 mm in diameter) of the wild-type strain and mutants were transferred to PDA plates and incubated in the dark at 25 °C. The assays for vegetative growth and conidial production were performed as described[48]. Mycelia were collected by suction filtration from liquid cultures that had been grown in 250 ml PDB and shaken at 180 rpm at 25 °C for 7 days; the mycelia were dried at 50 °C and weighed. For conidial germination and appressorial formation, droplets of 10 μl of conidial suspensions ($1 \times 10^5$ conidia/ml) were dropped onto plastic coverslips and incubated at 25 °C for 24 h. Conidial germination and appressorial formation were measured. The percentages of conidial germination and appressorial formation were determined by microscopic examination of at least 100 conidia or appressoria. ROS stress response assays were carried out in PDA plates supplemented with 2 mM menadione and 3 mM $H_2O_2$, separately, and incubated in the dark at 25 °C for 7 days. Each test was repeated three times.

**Pathogenicity and plant infection assays.** To test virulence, 5 ml conidial suspension was evenly sprayed onto 4-week-old seedlings of susceptible *A. thaliana* ecotype Col-0 with a sprayer as described previously[48]. Inoculated plants were incubated in a chamber at 25 °C under dark and saturated humidity conditions for 24 h, and then transferred to a growth chamber at 25 °C, with a light/dark photoperiod of 12/12 h. Lesion formation was examined at 7 days post-inoculation (dpi). Each test was repeated three times. Trypan blue staining was used to assess the extent of fungal colonization, inoculated leaf tissues were fixed and destained in a methanol:chloroform:glacial acetic acid (6:3:1) solution, rehydrated in water, stained with trypan blue[49], and viewed by light microscopy.

**qRT-PCR analysis.** Total RNA was extracted from fungal hyphae using TaKaRa RNAiso Plus (TaKaRa, Dalian, China) following manufacturer's instructions. For cDNA synthesis, TransScript® II One-Step gDNA Removal and cDNA Synthesis SuperMix (TransGen Biotech, Beijing, China) were used. Real-time quantitative PCR was performed in a CFX96™ Real Time PCR system (Bio-Rad, Hercules, CA) using TransStart® Tip Green qPCR SuperMix (TransGen Biotech, Beijing, China). Gene expression levels were calculated relative to β-tubulin gene (Ch063_01222) using the $2^{-\Delta\Delta Ct}$ method[50].

**Fluorescence microscopy.** To detect the subcellular localization of the fusion proteins, hyphae of complementation transformants expressing GFP were observed under a Leica SP8 laser-scanning confocal microscope (LSCM) (Leica Microsystems, Wetzlar, Germany) with excitation/emission wavelengths of 488 nm/505–530 nm to detect GFP signals.

To mark the mitochondria, the MTS-GFP construct was constructed as previously described[17]. GFP was fused to the N terminus of the mitochondrial targeting signal and expressed in the deletion mutants. Green fluorescence in the vegetative hyphae was observed under a Leica SP8 LSCM.

*C. higginsianum* cells expressing fluorescent fusion proteins were grown for 2 days in CM or starved in MM-N. To label mitochondria, hyphae of GFP-harboring strains in selection medium were incubated in 100 nM MitoFluor Red 589 (Molecular Probes, M22424) for 2 min at room temperature. To label lipid droplets, hyphae of these strains in selection medium were incubated in Nile red solution consisting of 50 mM Tris/maleate buffer (pH 7.5) and 2.5 mg/ml Nile red Oxazone (9-diethylamino-5H-benzo-a-phenoxazine-5-one, Sigma).

**Transmission electron microscopy.** For TEM observation, conidia were fixed in fixative solution (Servicebio, Wuhan, China) for 12 h, and then dehydrated by

ethanol concentrations of 25, 50, 75, 90 and 100% for 10 min at each step. The samples were incubated in an absolute acetone:final Spurr resin mixture solution (1:1) for 1 h, transferred to absolute acetone:final Spurr resin mixture solution (1:3) for 3 h, and incubated overnight in final Spurr resin mixture. The samples were embedded, heated at 70 °C for 9 h, and cut into sections using a microtome. The sections were stained with uranyl acetate, followed by alkaline lead citrate, for 15 min each and observed by TEM (Hitachi H-7650).

**Measurement of mitochondrial transmembrane potential.** To measure the change in mitochondrial transmembrane potential (ΔΨm), strains were grown in MM medium for 3 days, washed with phosphate-buffered saline (PBS) three times, and mitochondrial transmembrane potential (ΔΨm) was measured using a mitochondrial membrane potential assay kit with JC-1 (Beyotime, C2006) according to the manufacturer's instructions. After incubation, the samples were washed twice with PBS, and the mean values of fluorescence intensities of the JC-1 monomers (green fluorescence emission at 525 nm) and JC-1 aggregates (red fluorescence emission at 595 nm) were measured using a Leica SP8 LSCM (Leica Microsystems, Wetzlar, Germany). At high concentrations (reflecting a high ΔΨ) JC-1 forms aggregates, displaying red fluorescence. At lower concentrations, when ΔΨ decreases or collapses, JC-1 is present as monomers, exhibiting green fluorescence. To calculate the green/red fluorescence ratio of JC-1, all images were operated with the same settings and pixel numbers were quantified for green or red fluorescence using ImageJ program.

**Mitophagy monitoring assays.** For mitophagy monitoring assays, the MTS-GFP-expressing strain was first grown in complete medium (CM) (1 g/L yeast extract, 6 g/L casein hydrolysate, and 10 g/L sucrose) for 2 days, and then incubated in basal medium with glycerol (BG; 1.6 g/L yeast nitrogen base, 2 g/L asparagine, 1 g/L $NH_4NO_3$, and 1.5% v/v glycerol) for 30 h. The mycelia were then starved by 6 h incubation in nitrogen-starvation minimal medium (MN) containing 2 mM PMSF (0.5 g/L KCl, 0.5 g/L MgSO4, 1.5 g/L $KH_2PO_4$, 0.1% (v/v) trace elements, 10 g/L glucose, pH 6.5)[17]. The samples were staining with 10 μM Cell Tracker Blue CMAC Dye (Molecular Probes, C2110) for 30 min at 25 °C before observation. For autophagy monitoring assays, CM-cultured mycelia were washed and incubated in MM-N for 6 h to induce nonselective autophagy.

**Yeast two-hybrid assays.** Yeast two-hybrid (Y2H) analysis was carried out using the Matchmaker® Gold Yeast Two-Hybrid System (Takara Clontech). Details of the primers used for the constructs used in Y2H are listed in Supplementary Data 2. Yeast strains carrying pGBKT7-P53/pGADT7-T or pGBD/pGAD were used as positive and negative controls, respectively. Yeast transformants expressing plasmids which harbor the indicated constructs were assayed for growth on nonselective plates (SD/–Leu/–Trp) and selective plates (SD/–Leu/–Trp/–His) with 1 mM X-gal for 3 days.

**Co-immunoprecipitation assays.** Co-immunoprecipitation (Co-IP) assays were used to confirm the protein interactions in vivo. *ChPhb1* cDNA was cloned into pNeo3300-FLAG, while *ChPhb2* cDNA was cloned into pNeo3300-GFP. The resulting vectors (ChPhb1-3×FLAG and ChPhb2-GFP) were co-transformed into the wild-type strain. Total proteins were isolated from transformants expressing both ChPhb1-3×FLAG and ChPhb2-GFP, and incubated with anti-Flag M2 affinity resins (Sigma Aldrich). Proteins that bound to the M2 resins were eluted after a series of washing steps following the manufacturer's instructions. Immunoblot analysis was performed with anti-FLAG (1:2000 dilution; Sigma Aldrich, USA) and anti-GFP (1:10000 dilution; TransGen Biotech, China) primary antibodies. Immunoblot signals were detected using the ChemiDoc XRS + system (Bio-Rad). Interactions of ChPhb1-3 × FLAG/ChPhb2-GFP, ChPhb1-3 × FLAG/ChATG24-GFP and ChATG24-3 × FLAG/ChPhb2-GFP were also measured based on the above methods. Similar methods were used to test the deletion strain *ΔChPhb1* expressing ChATG24-3 × FLAG with ChPhb2-GFP fusion constructs, and deletion strain *ΔChPhb2* expressing ChPhb1-3 × FLAG with ChATG24-GFP fusion constructs.

**Protein extraction and immunoblot analysis.** For immunoblot analysis, total proteins were isolated from vegetative hyphae as described[25], and then were separated by 12% SDS-PAGE and transferred to a PVDF membrane (Millipore, Ontario, Canada) with a Bio-Rad electroblotting apparatus. Immunoblot signals were measured using the ChemiDoc XRS + system (Bio-Rad).

**Bimolecular fluorescence complementation assays.** BiFC assays were conducted to confirm protein interactions in vivo. The ChPhb1[CYFP] and ChATG24[CYFP] fusion constructs were generated by cloning the *ChPhb1* and *ChATG24* fragments, respectively, into pNeo3300-CYFP. The ChPhb2[NYFP] and ChATG24[NYFP] fusion constructs were generated respectively by cloning the *ChPhb2* or *ChATG24* fragment, into pNeo3300-NYFP. The construct pairs ChPhb1[CYFP]/ChPhb2[NYFP], ChPhb1[CYFP]/ChATG24[NYFP], or ChPhb2[NYFP]/ChATG24[CYFP] were co-transformed into wild-type Ch-1 using ATMT, with NYFP- and CYFP-expressing strains used as negative controls. Transformants that were resistant to hygromycin

and neomycin were isolated and confirmed by PCR. YFP signals were examined under a Leica SP8 LSCM.

**Bioinformatics**. The full-length sequences of *ChPhb1* and *ChPhb2* were downloaded from the genomic database (http://fungi.ensembl.org/Colletotrichum_higginsianum/Info/Index) of *C. higginsianum* isolate IMI349063. Protein domain and motif predictions were performed with DOG (Domain Graph, version 1.0) software. Homologs to the ChPhb1 and ChPhb2 protein sequences from different organisms were identified from the GenBank database using the BLASTP algorithm. A phylogenetic tree was generated with MEGA software (version 7.0, http://www.megasoftware.net/index.php) using the neighbor-joining (NJ) algorithm.

**Statistics and reproducibility**. All values and error bars in graphs are means ± standard error of the mean (SEM); respective *n* values are indicated in Figure legends. Statistical analysis was calculated by two-tailed Student's *t* tests using the Graphpad Prism 8 software.

**Reporting summary**. Further information on research design is available in the Nature Research Reporting Summary linked to this article.

## Data availability
The source data underlying the graphs and real-time PCR analysis within the paper can be found in Supplementary Data 1. Unprocessed blots can be found in Supplementary Figs. 7–11. All other data, vectors and plasmids are available from the corresponding authors on reasonable request.

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

## Acknowledgements

We are grateful to Prof. Youliang Peng (China Agricultural University, China) for providing suggestions. This work was supported by grants from the National Natural Science Foundation of China (No. 31101399).

## Author contributions

YY conducted most of the experiments and wrote the manuscript. TJ, YQ, LC, CX and LH conducted parts of the research. HJ and BC gave critical suggestions for structure and writing of the manuscript. HT revised the manuscript. ZL supervised all aspects of the study, reviewed critically, and revised the manuscript. All authors read and approved the contents of this manuscript.

## Competing interests

The authors declare no competing interests.
