## [Peer Review File · Communications Biology]

Reviewers' comments:

Reviewer #1 (Remarks to the Author):

This manuscript by Yan et al. described the roles of the prohibitin proteins ChPhb1 and ChPhb2 in the function of mitochondria, mitophagy and virulence in *C. higginsianum*. The authors found that ChPhb1 and ChPhb2 interact with each other, and both ChPhb1 and ChPhb2 interact with the mitophagic protein ChATG24 in mitochondria. They also found that ChATG24 is important for virulence, and deletion of either ChPhb1 or ChPhb2 results in mislocalization of ChATG24. Overall data are rich. Some data are reasonably explained, but some data are not clearly interpreted. Additional data and more careful interpretation/explanation are required. Comments are detailed below.

Major comments:

1. In the Figure 2b, Co-IP assay showed that ChPhb1 interacts with ChPhb2 in vivo. Indeed, the signal of ChPhb2-GFP was too weak. Also, in the Figure 6b, Co-IP assay showed that ChPhb1 interacts with ChATG24, but the signal of ChATG24-GFP was quite weak. Is it possible that the interactions between them are unstable? Did you test whether deletion of ChATG24 affected the interaction between ChPhb1 and ChPhb2? And whether deletion of ChPhb2 or ChPhb1 affected the interactions between ChPhb1 or ChPhb2 and ChATG24?
2. In the Figure 3a-3d, the authors described that deletion of ChPhb2 displayed defective growth and conidiation, but deletion of ChPhb1 did not. In addition, the ChPhb1ChPhb2 double deletion mutant showed a similar phenotype with ChPhb2 deletion mutant. The current statistical comparison of ChPhb2 deletion is with wild-type strain. Lack of the statistical comparison with ChPhb1ChPhb2 double deletion strain is a problem. Statistical differences between ChPhb2 single deletion and ChPhb1ChPhb2 double deletion strain should be performed.
3. Again, in the Figure 3h and Figure 4c-4d, lack of the statistical comparison of single deletion with double deletion is a big problem. In the page 15 lines 339, the authors concluded that ChPhb2, but not ChPhb1, is involved in the vegetative growth and conidiation. However, in the assays of mitochondrial transmembrane potential and menadione sensitivity, it seems that the double deletion strain showed more serious phenotypes than their single deletion strain. How to explain the differences in the phenotypes, including growth, conidiation, mitochondrial transmembrane potential, and menadione sensitivity between ChPhb2 single deletion strain and ChPhb1ChPhb2 double deletion strain? What is the functional relationship between ChPhb1 and ChPhb2?
4. In the Figure 4d, the authors examined the menadione sensitivity of the strains to evaluate the accumulation of ROS. For a much more convincing demonstration, the ROS levels in these strains should be measured.
5. In the page 17 lines 395-396, the authors claim that "ChPhb1 and ChPhb2 deletion mutants showed conspicuously increased sensitivity to mitochondrial ROS, but not to exogenous H₂O₂.", however, the authors did not show the results of sensitivity to exogenous H₂O₂ throughout the manuscript.

Minor comments

1. In the page 6 line 131, the authors claim that "while the constructs ChPhb1-YFPCTF and YFPNTF were co-transformed into strain Ch-1 as a negative control", the "ChPhb1-YFPCTF" should be "YFPCTF"?
2. In the Figure 5, page 10 line 211, the authors described the results of ChPhb1 or ChPhb2 single deletion strain, but did not describe the results of the double deletion strain, although the Figure 5 showed the results of double deletion strain.

Reviewer #2 (Remarks to the Author):

The manuscript reports on the analysis of two prohibitin homologs in the plant pathogenic fungus

Colletotrichum higginsianum. Mutation of one of the two copies impacts mitochondrial function, the mitochondrial-specific aspect of autophagy, and virulence on plants. The authors then link the mitophagy effect to a physical interaction between the Phb proteins with the autophagy protein Atg24.

As the first study to explore the role of prohibitins in pathogenic fungi, the research will be of interest to those exploring the genetic basis of plant disease in fungi, autophagy, and links between specific organelle functions and pathogenicity.

Major points:

A number of strains and likely experiments were created and performed, but then the information does not appear in the body of the main text. Examples are

- Lines 118-121 GFP strain; are these constructions able to complement the mutant phenotypes?
- The complemented strains (described in the Methods lines 469-476) should be discussed after line 164.
- Lines 192-198: mention the other stresses tested. If specific to mitochondria, there should be the impact with menadione but potentially not with other ROS. This is mentioned on lines 393-397 but could appear in the Results section and with more detail about which ROS were tested.

One question on evolution is around the functions of Phb1 and Phb2. Is there any information about how these proteins function as a complex? To elaborate, the authors present evidence for Phb1-Phb2 interactions, but phenotypes are only observed for the *phb2* deletion strain. It could be that prohibitin functions as a dimer (or likely in a complex), presumably here as a heterodimer, but it would have been interesting to use one of the quicker assays (e.g. Y2H) to test if there are also Phb2-Phb2 interactions to form homodimers. This could then be used to explain why there is no need for Phb1.

Minor points:

Line 31: check language, i.e. duplication of 'regulate'.

Line 108: write out 'Y2H' in full first time used.

Line 128: write out 'BiFC' in full first time used.

Line 177: write out 'TEM' in full first time used.

Figure S5: the G in 'GFP-ATG8' has been cut off.

Double check reference formatting, e.g. lines 637, 643 and 659 no . after Cell; line 645 spelling 'Ccontrol'; line 672 italics on 'RCH1'.

Reviewer #3 (Remarks to the Author):

Yan et al sought to explore the regulatory mechanisms involved in mitophagy in filamentous fungi, using the plant pathogenic fungus *Colletotrichum higginsianum* as model. This study demonstrated that mitochondrial prohibitin is required for organellar function and mitophagy. The conclusion is original in this field. These findings broaden the current understandings of mitophagy in fungi and attractive to researchers in this discipline. The manuscript is well organized and written, which deserves publication in this journal. There are some suggestions for improving manuscript.

So, my suggestion is Major revision.

Further concerns:

1. Line 87. Not use "regulating". Actually, it is a biochemical pathway of the prohibitin-mediated translocation of Atg24, which ultimately is involved in fungal virulence.
2. Line 68-70. Revise this sentence to make the mean more precise. The role of mitophagy in fungal pathogen has been well revealed. The regulatory mechanisms are really not fully explored.
3. Line 213-219. There are significant conflicts between two experiments. As revealed in line 209-212,

Phb1 and Phb2 are involved in mitophagy under nitrogen-starvation medium. However, hereafter, fungal strains were stressed in MN medium, also a nitrogen-starvation minimal medium (Lin563). GFP-signals were detected in vacuoles of the gene disruption mutants. WHY, two results from nitrogen-starvation conditions are different.

Another problem is that this part is not absolutely necessary. As revealed above, Phb1 and Phb2 localize in mitochondria and might functions as a receptor, due to its interaction with Atg24 (line221,272). So, it is unnecessary to detect the total autophagic flux. However, it is suggested to add additional experiments. At least, you should explore whether Atg8 and Atg11 are necessary for mitophagy in *Colletotrichum higginsianum*.

Journal: Communications Biology

Manuscript number: COMMSBIO-21-3287-T

RESPONSES TO COMMENTS

Reviewer: 1

This manuscript by Yan et al. described the roles of the prohibitin proteins ChPhb1 and ChPhb2 in the function of mitochondria, mitophagy and virulence in *C. higginsianum*. The authors found that ChPhb1 and ChPhb2 interact with each other, and both ChPhb1 and ChPhb2 interact with the mitophagic protein ChATG24 in mitochondria. They also found that ChATG24 is important for virulence, and deletion of either ChPhb1 or ChPhb2 results in mislocalization of ChATG24. Overall data are rich. Some data are reasonably explained, but some data are not clearly interpreted. Additional data and more careful interpretation/explanation are required. Comments are detailed below.

Response: Thanks for your comments. Additional data have been added into the manuscript and your insightful comments have been addressed below.

Comment 1: In the Figure 2b, Co-IP assay showed that ChPhb1 interacts with ChPhb2 *in vivo*. Indeed, the signal of ChPhb2-GFP was too weak. Also, in the Figure 6b, Co-IP assay showed that ChPhb1 interacts with ChATG24, but the signal of ChATG24-GFP was quite weak. Is it possible that the interactions between them are unstable? Did you test whether deletion of ChATG24 affected the interaction between ChPhb1 and ChPhb2? And whether deletion of ChPhb2 or ChPhb1 affected the interactions between ChPhb1 or ChPhb2 and ChATG24?

Response: As suggested, Co-IP assays have been carried out again to confirm the interactions between ChPhb1 and ChPhb2, and between ChPhb2 and ChAtg24. Please see Figs. 3b and 6b. The results showed that signals of both assays were strong and clear, indicating the interactions among them. Furthermore, results of yeast two-hybrid and BiFC also demonstrated the stable interactions of ChPhb1/ChPhb2, ChPhb1/ChAtg24, and ChPhb2/ChAtg24 *in vitro* and *in vivo*.

As suggested, we also have performed Co-IP assays in separate deletion mutants of *ChPhb1*, *ChPhb2* or *ChAtg24* to confirm the interactions between the proteins. The results demonstrated that deleting any one of those three genes did not affect the

interaction between the other two. Please see Fig. 6d. The results are described as follows: “To further confirm the interactions between ChPhb1 and ChPhb2, ChPhb1 and ChAtg24, or ChPhb2 and ChAtg24 in the absence of *ChATG24*, *ChPhb2* or *ChPhb1*, three fusion construct sets of *ChPhb1-3×FLAG* and *ChPhb1-GFP*, *ChPhb1-3×FLAG* and *ChATG24-GFP*, or *ChATG24-3×FLAG* and *ChPhb2-GFP* were separately transformed into the $\Delta ChATG24$, $\Delta ChPhb2$ or $\Delta ChPhb1$ mutant. Results of Co-IP assays showed that ChPhb1 interacted with ChPhb2 in the $\Delta ChATG24$ mutant, ChPhb1 interacted with ChAtg24 in the $\Delta ChPhb2$ mutant, and ChPhb2 interacted with ChAtg24 in the $\Delta ChPhb1$ mutant (Fig. 6d), suggesting that the absence of any one of these three proteins did not affect the interactions between the other two.” (lines 264-272)

Comment 2: In the Figure 3a-3d, the authors described that deletion of ChPhb2 displayed defective growth and conidiation, but deletion of ChPhb1 did not. In addition, the ChPhb1ChPhb2 double deletion mutant showed a similar phenotype with ChPhb2 deletion mutant. The current statistical comparison of ChPhb2 deletion is with wild-type strain. Lack of the statistical comparison with ChPhb1ChPhb2 double deletion strain is a problem. Statistical differences between ChPhb2 single deletion and ChPhb1ChPhb2 double deletion strain should be performed.

Response: As suggested, statistical comparisons have been conducted for all tested strains using LSD at $P=0.05$. The analyses indicated that there were no significant differences in mycelial growth or conidiation between $\Delta ChPhb2$ and $\Delta ChPhb1/ChPhb2$. Please see the revised Fig. 2.

Comment 3: Again, in the Figure 3h and Figure 4c-4d, lack of the statistical comparison of single deletion with double deletion is a big problem. In the page 15 lines 339, the authors concluded that ChPhb2, but not ChPhb1, is involved in the vegetative growth and conidiation. However, in the assays of mitochondrial transmembrane potential and menadione sensitivity, it seems that the double deletion strain showed more serious phenotypes than their single deletion strain. How to explain the differences in the phenotypes, including growth, conidiation, mitochondrial transmembrane potential, and menadione sensitivity between ChPhb2 single deletion strain and ChPhb1ChPhb2 double deletion strain? What is the functional relationship between ChPhb1 and ChPhb2?

Response: As suggested, statistical comparison have been conducted for all tested strains using LSD at P=0.05. Please see the revised Figs. 2 & 4.

In this study, we found that there were no significant differences between $\Delta ChPhb2$ and $\Delta ChPhb1/ChPhb2$ mutants in vegetative growth, conidiation or pathogenicity. However, the $\Delta ChPhb1/ChPhb2$ deletion mutant showed significant damage in mitochondrial transmembrane potential and reduced menadione sensitivity compared to $\Delta ChPhb2$ mutant. The degree of mitochondrial damage caused by deletion of *ChPhb1*, *ChPhb2*, or *ChPhb1/ChPhb2* did not correspond to their respective vegetative growth or conidiation. These indicated that *ChPhb1* and *ChPhb2* functioned not only individually but also collaboratively in mitochondria. Previous studies showed that Phb1 and Phb2 function as a heterodimeric complex (Wei et al., 2017; Hernando-Rodríguez et al., 2018). To further confirm the role of ChPhb1 and ChPhb2, yeast two-hybrid assays of ChPhb1 or ChPhb2 self-interaction was assessed and no self-interaction was found for ChPhb1 or ChPhb2 (Please see Fig. 3a). These results indicated that ChPhb1 and ChPhb2 presumably formed a heterodimeric complex. This could explain why ChPhb1 and ChPhb2 function independently. Please see the the discussion section in lines 378-390.

Comment 4: In the Figure 4d, the authors examined the menadione sensitivity of the strains to evaluate the accumulation of ROS. For a much more convincing demonstration, the ROS levels in these strains should be measured.

Response: We have tried many times to measure mitochondrial ROS levels in these strains using DCFH-DA and mitoSOX-red. However, we cannot obtain stable data neither detecting directly nor detecting with isolated mitochondria in *C. higginsianum*. Therefore, we have added this text to the discussion: "We attempted many times to measure mitochondrial ROS levels levels in these strains using DCFH-DA and mitoSOX-red. However, we were unable to obtain consistent data either when attempting to directly detect *in situ* or when using isolated mitochondria of *C. higginsianum*. Future work is needed to develop a more efficient method of ROS detection in mitochondria of *C. higginsianum*." (lines 420-424)

Comment 5: In the page 17 lines 395-396, the authors claim that "ChPhb1 and ChPhb2 deletion mutants showed conspicuously increased sensitivity to mitochondrial ROS, but not to exogenous H₂O₂", however, the authors did not show

the results of sensitivity to exogenous H₂O₂ throughout the manuscript.

Response: As suggested, in the results section, we have added the description of the the results of sensitivity to exogenous H₂O₂, and the data are shown in Extended Data Fig. 4b. These results are described as follows: “No significant changes were observed between $\Delta ChPhb1$, $\Delta ChPhb2$ or $\Delta ChPhb1/ChPhb2$ mutant and the wild type on the plates amended with H₂O₂ (Extended Data Fig. 4b).” in lines 210-213.

Comment 6: In the page 6 line 131, the authors claim that “while the constructs ChPhb1-YFPCTF and YFPNTF were co-transformed into strain Ch-1 as a negative control”, the “ChPhb1-YFPCTF” should be “YFPCTF”?

Response: As suggested, we have revised “ChPhb1-YFP^{CTF}” as “YFP^{CTF}”.

Comment 7: In the Figure 5, page 10 line 211, the authors described the results of ChPhb1 or ChPhb2 single deletion strain, but did not describe the results of the double deletion strain, although the Figure 5 showed the results of double deletion strain.

Response: As suggested, we have added the description of results of double deletion strain. The results are described as “When the mycelia were transferred from glycerol medium to nitrogen-starvation medium, GFP signals were clearly observed in mycelial vacuoles of the wild type but not of the $\Delta ChPhb1$, $\Delta ChPhb2$ or $\Delta ChPhb1/ChPhb2$ mutants (Fig. 5)”. (lines 224-227).

Reviewer: 2

The manuscript reports on the analysis of two prohibitin homologs in the plant pathogenic fungus *Colletotrichum higginsianum*. Mutation of one of the two copies impacts mitochondrial function, the mitochondrial-specific aspect of autophagy, and virulence on plants. The authors then link the mitophagy effect to a physical interaction between the Phb proteins with the autophagy protein Atg24. As the first study to explore the role of prohibitins in pathogenic fungi, the research will be of interest to those exploring the genetic basis of plant disease in fungi, autophagy, and links between specific organelle functions and pathogenicity.

Response: Thank you for your positive comments. Additional data have been added into the manuscript and your insightful comments have been addressed below.

Comment 1: A number of strains and likely experiments were created and performed, but then the information does not appear in the body of the main text. Examples are
- Lines 118-121 GFP strain; are these constructions able to complement the mutant phenotypes?

Response: Yes, the complementation strains can complement the phenotypes of mutant. We have added the description as “The complementation strains which complement the phenotypes of deletion mutants were obtained (Fig. 2 and Extended Data Fig. 1)”. For the data of the complementation strains, please see Fig. 2 and Extended Data Fig. 1.

Comment 2: - The complemented strains (described in the Methods lines 469-476) should be discussed after line 164.

Response: As suggested, we have added the information of the complemented strains into the results section “*ChPhb1 and ChPhb2 are required for full virulence*”. Please see the lines 107-114, 125-134 for details.

Comment 3: - Lines 192-198: mention the other stresses tested. If specific to mitochondria, there should be the impact with menadione but potentially not with other ROS. This is mentioned on lines 393-397 but could appear in the Results section and with more detail about which ROS were tested.

Response: As suggested, in the results section, we have added the description of the the results of sensitivity to exogenous H₂O₂, and the data are shown in Extended Data Fig. 4b. These results are described as follows: “No significant changes were observed between $\Delta ChPhb1$, $\Delta ChPhb2$ or $\Delta ChPhb1/ChPhb2$ mutant and the wild type on the plates amended with H₂O₂ (Extended Data Fig. 4b)” in lines 211-214.

Comment 4: One question on evolution is around the functions of Phb1 and Phb2. Is there any information about how these proteins function as a complex? To elaborate, the authors present evidence for Phb1-Phb2 interactions, but phenotypes are only observed for the phb2 deletion strain. It could be that prohibitin functions as a dimer (or likely in a complex), presumably here as a heterodimer, but it would have been interesting to use one of the quicker assays (e.g. Y2H) to test if there are also Phb2-Phb2 interactions to form homodimers. This could then be used to explain why

there is no need for Phb1.

Response: As suggested, yeast two-hybrid assays of ChPhb1 or ChPhb2 self-interaction were assessed and no self-interaction was found for ChPhb1 or ChPhb2 (Please see Fig. 3a). These results indicated that ChPhb1 and ChPhb2 might form a heterodimeric complex. The results are described as “Previous studies in mammals showed that Phb1 and Phb2 can interact as a heterodimeric complex (Wei et al., 2017; Hernando-Rodríguez et al., 2018). To further confirm the role of ChPhb1 and ChPhb2, yeast two-hybrid assays of ChPhb1 or ChPhb2 self-interaction were assessed and no self-interaction was found for ChPhb1 or ChPhb2 (Fig. 3a). These results indicated that ChPhb1 and ChPhb2 might form as a heterodimeric complex.” (lines 145-149).

Comment 5: Line 31: check language, i.e. duplication of ‘regulate’.

Response: Corrected as suggested.

Comment 6: Line 108: write out ‘Y2H’ in full first time used.

Response: As suggested, we have reworded ‘Y2H’ as ‘yeast two-hybrid (Y2H)’.

Comment 7: Line 128: write out ‘BiFC’ in full first time used.

Response: Done as suggested, we have reworded ‘BiFC’ as ‘bimolecular fluorescence complementation (BiFC)’.

Comment 8: Line 177: write out ‘TEM’ in full first time used.

Response: As suggested, we have reworded ‘TEM’ as ‘transmission electron microscopy (TEM)’.

Comment 9: Figure S5: the G in ‘GFP-ATG8’ has been cut off.

Response: Corrected as suggested.

Comment 10: Double check reference formatting, e.g. lines 637, 643 and 659 no. after Cell; line 645 spelling ‘Ccontrol’; line 672 italics on ‘RCH1’.

Response: As suggested, we have checked carefully for the reference formatting and revised all formatting problems. Please see the revised references section for details.

Reviewer: 3

Yan et al sought to explore the regulatory mechanisms involved in mitophagy in filamentous fungi, using the plant pathogenic fungus *Colletotrichum higginsianum* as model. This study demonstrated that mitochondrial prohibitin is required for organellar function and mitophagy. The conclusion is original in this field. These findings broaden the current understandings of mitophagy in fungi and attractive to researchers in this discipline. The manuscript is well organized and written, which deserves publication in this journal. There are some suggestions for improving manuscript. So, my suggestion is Major revision.

Response: Thank you for your positive comments. Additional data have been added into the manuscript and your insightful comments have been addressed below.

Comment 1: Line 87. Not use “regulating”. Actually, it is a biochemical pathway of the prohibitin-mediated translocation of Atg24, which ultimately is involved in fungal virulence.

Response: As suggested, we have revised “regulating” as “affecting”.

Comment 2: Line 68-70. Revise this sentence to make the mean more precise. The role of mitophagy in fungal pathogen has been well revealed. The regulatory mechanisms are really not fully explored.

Response: As suggested, we have revised this sentence to “However, the regulatory mechanisms of mitophagy in fungal pathogenesis are not well understood.”

Comment 3: Line 213-219. There are significant conflicts between two experiments. As revealed in line 209-212, Phb1 and Phb2 are involved in mitophagy under nitrogen-starvation medium. However, hereafter, fungal strains were stressed in MN medium, also a nitrogen-starvation minimal medium (Lin563). GFP-signals were detected in vacuoles of the gene disruption mutants. WHY, two results from nitrogen-starvation conditions are different.

Response: The fluorescent labels and experimental protocols were both different between mitophagy and autophagy monitoring assays. For mitophagy monitoring assays, the MTS-GFP-expressing strain was incubated in basal medium with glycerol to induce proliferation of mitochondrial, and then starved by culturing in MN medium.

For autophagy monitoring assays, the ATG8-GFP-expressing strain was just incubated in MN medium. To avoid confusion, we have added the methods of autophagy monitoring assays in the Methods section. These methods have been described as “For autophagy monitoring assays, CM-grown mycelia were washed and subjected to nitrogen starvation (cultured in MM-N for 6 h) to induce nonselective autophagy”. (lines 618-621)

Comment 4: Another problem is that this part is not absolutely necessary. As revealed above, Phb1 and Phb2 localize in mitochondria and might functions as a receptor, due to its interaction with Atg24 (line221,272). So, it is unnecessary to detect the total autophagic flux. However, it is suggested to add additional experiments. At least, you should explore whether Atg8 and Atg11 are necessary for mitophagy in *Colletotrichum higginsianum*.

Response: From the mitophagy and autophagy assays, we could conclude that *ChPhb1* and *ChPhb2* were essential for mitophagy, but not all types of autophagy.

As suggested, we have explored whether Atg8 is necessary for mitophagy in *C. higginsianum*. The results are described as follows (lines 238-242): “Since previous studies reported that ATG8 is essential for all types of autophagy, we generated a $\Delta ChATG8$ deletion mutant carrying Mito-GFP. The results showed that no vacuolar GFP signal was observed in the $\Delta ChATG8$ mutant (Fig. 5), suggesting that delivery of Mito-GFP to the vacuole in *C. higginsianum* indeed required *ChATG8*.”. Please see Fig. 5 for details.

REVIEWERS' COMMENTS:

Reviewer #1 (Remarks to the Author):

This manuscript has been substantially modified by authors, following the comments of reviewers of the original version.

Concerning my comments to the original version, most of the comments have been properly addressed. However, they failed to measure the ROS levels in *C. higginsianum* unfortunately, but the conclusion of this work is mostly not affected. I have no more questions on this revised manuscript.

Reviewer #2 (Remarks to the Author):

My feeling is that the authors have addressed the points raised by the reviewers, certainly those made by myself.

Reviewer #3 (Remarks to the Author):

My concerns have been addressed, and the manuscript is acceptable as it is.

Journal: Communications Biology

Manuscript number: COMMSBIO-21-3287

RESPONSE TO REFEREES

Reviewer #1:

This manuscript has been substantially modified by authors, following the comments of reviewers of the original version. Concerning my comments to the original version, most of the comments have been properly addressed. However, they failed to measure the ROS levels in *C. higginsianum* unfortunately, but the conclusion of this work is mostly not affected. I have no more questions on this revised manuscript.

Response: Thanks for reviewer's recognition.

Reviewer #2:

My feeling is that the authors have addressed the points raised by the reviewers, certainly those made by myself.

Response: Thanks for reviewer's recognition.

Reviewer #3:

My concerns have been addressed, and the manuscript is acceptable as it is.

Response: Thanks for reviewer's recognition.